# Towards Biologically Plausible Online Hebbian Learning: Two-Timescale Local Rules for Spiking Neural Brain Interfaces

## Abstract

Brain-Computer Interfaces face neural signal instability and tight memory budgets in real-time implantable settings. We introduce an online intracortical SNN decoder trained with a temporally local, layer-local but spatially non-local three-factor rule with dual-timescale eligibility traces, avoiding backpropagation through time and requiring memory that is constant in sequence length. On two primate datasets, this Online SNN attains Pearson correlations of at least $R \geq 0.63$ on Zenodo Indy and $R \geq 0.81$ on MC Maze while converging faster in early training than a BPTT-trained SNN and reducing measured training memory by 63–86% at sequence length $T = 1000$. The learning rule combines synapse-specific dual-timescale Hebbian accumulators, error-modulated updates, and integer-friendly RMS homeostasis, and operates without unrolled computational graphs or adaptive optimizer state. Closed-loop simulations with synthetic neural populations demonstrate online supervised adaptation to neural disruptions and learning from scratch without offline calibration. Overall, the method provides a memory-efficient, continuously adaptive decoder that is temporally local and Hebbian but still relies on spatial backpropagation across layers, yielding a partially biologically plausible algorithm that is competitive with BPTT-trained SNN and Kalman baselines while trading some offline accuracy relative to LSTM/GRU decoders for online learning and deployment-oriented properties.

## 1 Introduction

Brain–Computer Interfaces (BCIs) translate neural activity into control signals, bypassing conventional neuromuscular pathways (Pandarinath et al., 2017; Shih et al., 2012; Wolpaw et al., 2002; Collinger et al., 2013; Bouton et al., 2016). Invasive approaches provide high-fidelity recordings but face barriers including signal instability, noise, and resource constraints, motivating adaptive, efficient, and biologically grounded decoders (Furber et al., 2014; Roy et al., 2019; Taeckens & Shah, 2024; Hussain et al., 2015; Bhaduri et al., 2018; Maynard et al., 1997; Woeppel et al., 2021). However, traditional methods struggle with non-stationarities while deep models require frequent recalibration (Wu et al., 2004; Huang et al., 2021; Jiang et al., 2018; Sussillo et al., 2016) (Shaikh et al., 2019). In this work we focus specifically on intracortical Utah-array BCIs and on deployment-oriented properties—constant-in-$T$ training memory, per-timestep supervised updates, and integer-friendly implementations—rather than proposing a universal replacement for all temporally local SNN learning rules.

### 1.1 Challenges and Motivation

Neural recordings drift due to electrode encapsulation and neural plasticity, forcing frequent recalibration and motivating continuous online adaptation that does not interrupt use (Dangi et al., 2014; Ganguly & Carmena, 2009; Wolpaw et al., 2002; Sussillo et al., 2016; Woeppel et al., 2021). Electrophysiological data are noisy and high-dimensional, complicating low-latency decoding, although SNNs with sparse, event-driven activity can help manage these characteristics (Huang et al., 2021; Bhaduri et al., 2018). Models often fail to generalize across sessions or individuals without retraining, requiring adaptive mechanisms. Many high-performing decoders rely on backpropagation through

time and large batch updates that are ill-suited to power- and memory-constrained implants, and error backpropagation itself is considered biologically implausible due to the weight transport problem (Lillicrap et al., 2020) and on-chip learning constraints in neuromorphic systems (Basu et al., 2022), motivating BPTT-free rules that still admit practical gradient computation.

Our goal is to close a specific gap between offline high-accuracy BCI decoders and online adaptive but resource-intensive training schemes. Recurrent ANNs trained with BPTT achieve strong offline decoding but require storing long activation sequences and performing batched updates, which limits their suitability for long-running, implantable BCIs. Classical adaptive decoders such as Kalman filters or simple delta-rule updates provide online updates with constant memory but lack the flexibility of deep models in the face of complex, non-stationary intracortical signals. We therefore investigate whether a dual-timescale three-factor SNN learning rule with synapse-specific eligibility traces can provide a middle ground: training memory that is constant in sequence length, temporally local online updates, and competitive decoding relative to BPTT-trained SNNs and classical decoders, while accepting a controlled loss in offline accuracy relative to LSTM/GRU baselines.

We evaluate our approach in closed-loop settings because these are necessary for capturing the dynamic co-adaptation between neural activity and decoder learning—an interaction that does not emerge in static, open-loop (offline) analyses (Carmena, 2013; Dangi et al., 2014; Orsborn et al., 2014; Sadtler et al., 2014).

Beyond these general BCI considerations, SNNs are attractive in this setting for three concrete reasons. First, three-factor plasticity rules allow per-timestep supervised updates without storing past activations, enabling true online learning with memory that is constant in sequence length. Second, the event-driven, spike-based representation matches intracortical recordings and supports sparse computation, which is important for low-power deployment. Third, our implementation uses integer-friendly RMS normalization and lookup-table-based scaling rather than floating-point adaptive optimizers, aligning the learning rule with neuromorphic hardware constraints. These advantages come at a cost: offline decoding accuracy remains below that of LSTM/GRU baselines, but we show that the resulting trade-off is favorable for deployment scenarios that prioritize continuous adaptation and strict memory budgets.

Existing approaches often address these elements separately, lacking a unified mechanism that integrates multi-factor plasticity and two-timescale consolidation within an SNN framework. This paper introduces an *online adaptive SNN decoder* that combines these components. Our framework, featuring Hebbian updates at each synapse (chosen for their biological relevance and computational locality in time) and fast/slow weight consolidation (to address the stability-plasticity dilemma),

- avoids backpropagation through time, removing the $O(T)$ activation memory term and enabling training with memory that is constant in sequence length while retaining standard feedforward backpropagation across layers;
- enables per-timestep supervised online updates to track neural non-stationarities during continuous operation, reducing the need for explicit recalibration and supporting long-term BCI use;
- achieves competitive decoding accuracy relative to BPTT-trained SNNs and classical decoders on two primate intracortical datasets (MC Maze and Zenodo Indy), while using substantially less measured training memory and accepting a controlled loss in offline correlation compared to LSTM/GRU baselines;
- operates with a minimal training memory footprint and an integer-friendly, optimizer-free update suitable for real-time, implantable BCI hardware: the learning rule requires no unrolled backpropagation graphs or replay buffers, only short eligibility traces and running statistics, aligning with requirements for neuromorphic on-chip learning (Davies et al., 2018; Indiveri & Liu, 2015; Merolla et al., 2014).

In summary, our contribution complements two main classes of prior work. Relative to BPTT-based SNN training, we keep the same architectural expressiveness but replace time-unrolled gradient descent with a dual-timescale three-factor rule whose training memory is constant in sequence length and whose updates run in a true online mode. Relative to temporally local rules such as OTTT, S-TLLR, SLTT, and TESS, we focus on intracortical BCI decoding rather than large-scale computer vision benchmarks and rely on synapse-specific dual-timescale Hebbian accumulators with integer-friendly RMS homeostasis instead of factorized $O(N)$ traces, yielding a different

accuracy–adaptation–memory trade-off at Utah-array scales. Appendix E, Table 6 empirically compares these synapse-specific $O(N^2)$ traces to neuron-level $O(N)$ factorized traces on ten Zenodo sessions and finds roughly 0.06–0.08 higher correlation for the former, supporting this design choice at Utah-array scales.

## 1.2 Related Work

**Local three-factor SNN learning.** Classical spike-timing-dependent plasticity (STDP) uses pre- and postsynaptic spike timing for unsupervised learning but lacks task-specific supervision (Song et al., 2000; Caporale & Dan, 2008). Three-factor extensions incorporate modulatory signals: reward-modulated STDP uses sparse, delayed dopamine-like signals (Florian, 2007; Frémaux & Gerstner, 2016; Izhikevich, 2007), while eligibility trace methods like SuperSpike and e-prop use broadcast error signals with local traces (Zenke & Ganguli, 2018; Bellec et al., 2020). These approaches enable online learning without time-unrolled graphs but differ in trace derivation and error propagation mechanisms. More recent work such as online training through time (OTTT) and temporally local rules including S-TLLR, SLTT, and TESS (Xiao et al., 2022; Apolinario & Roy, 2025; Meng et al., 2023; Apolinario et al., 2025) further explore temporally and/or spatially local updates, predominantly for large-scale computer vision benchmarks. Our method instead focuses on intracortical BCI decoders with synapse-specific dual-timescale Hebbian accumulators and an optimizer-free, integer-friendly implementation. Hardware demonstrations of local STDP/three-factor rules include floating-gate and dendritic designs (Gopalakrishnan & Basu, 2017; Hussain et al., 2015).

**Surrogate gradient and BPTT SNN training.** Modern SNN training often employs surrogate gradients to handle spike non-differentiability, enabling backpropagation through time with strong supervision (Neftci et al., 2019). While suitable for offline training, BPTT requires O(T) memory for computational graph storage (T as in timesteps) and is considered biologically implausible due to weight transport requirements. Recent advances improve gradient flow and temporal alignment but maintain the fundamental BPTT framework.

**Adaptive and online BCI decoders.** Traditional BCI decoders like Kalman filters adapt via parameter updates but struggle with non-stationarities. Deep learning approaches achieve higher performance but require large amounts of data (Huang et al., 2021). Closed-loop adaptation and co-adaptive systems address decoder-user interaction but often rely on batch updates rather than continuous online learning (Carmena, 2013; Dangi et al., 2014; Orsborn et al., 2014; Bhaduri et al., 2018). From a hardware perspective, neuromorphic implementations favor local, on-chip learning rules (Basu et al., 2022).

**Positioning.** Our approach combines temporally local, layer-local three-factor learning with dense per-timestep error signals, dual-timescale eligibility traces, and hardware-friendly implementations. Unlike existing methods, we frame eligibility traces as *Hebbian accumulators modulated by reinforcement* rather than BPTT-approximating gradient surrogates, enabling continuous online adaptation with training memory that is constant in sequence length while remaining spatially non-local and therefore only partially biologically plausible.

## 2 Proposed SNN Architecture and Learning Algorithm

**Problem Setup and Notation.** We observe spike count vectors $\mathbf{x}_t \in \mathbb{R}^N$ and predict 2D velocity $\mathbf{y}_t \in \mathbb{R}^2$ at time bins $t = 1, \ldots, T$. Hidden layer $k$ has membrane potentials $\mathbf{u}_t^{(k)}$, spikes $\mathbf{s}_t^{(k)}$, with the first layer maintaining recurrent state $\mathbf{s}_{t-1}^{(1)}$. The final layer's membrane potentials directly represent velocity predictions: $\hat{\mathbf{y}}_t = \mathbf{u}_t^{(3)}$. We train online by minimizing per-timestep squared error $\mathcal{L}_t = \|\hat{\mathbf{y}}_t - \mathbf{y}_t\|_2^2$ using a local three-factor rule with dual eligibility traces. No unrolled computational graph or replay buffer is required. A complete summary of notation is provided in Appendix B.

## 2.1 Core Architecture

Our network is built around Leaky Integrate-and-Fire (LIF) neurons, chosen for their balance of biological realism and computational tractability (see Appendix F for background). Each neuron's

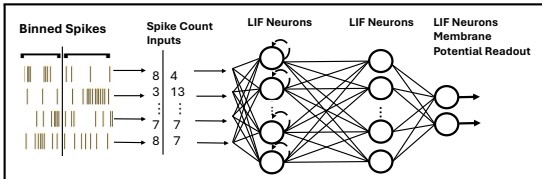

Figure 1: Proposed SNN architecture with recurrent connections and 2D velocity output, highlighting the three-layer LIF decoder with a recurrent first hidden layer used in all SNN experiments.

membrane potential $u_i(t)$ integrates incoming synaptic currents, leaks with a decay factor $\beta$, and emits a spike when reaching a threshold:

$$u_i(t+1) = \beta\, u_i(t) + I_i(t) - s_i(t).$$

Learning uses surrogate gradient methods (Neftci et al., 2019; Zenke & Vogels, 2021; Bellec et al., 2020) to handle spike non-differentiability, creating a neuronal sensitivity gate that focuses plasticity on neurons near threshold. Full derivations and homeostatic controls are in Appendix D. We use $d_{\mathrm{LIF}}$ as a per-timestep local gate, without temporal backpropagation (Algorithm 6).

The three-layer architecture (Figure 1) includes recurrent connections in the first hidden layer. Unlike baseline models that use spike rates, our SNN processes raw spike counts directly. Full details are in Appendix C.

## 2.2 Learning Algorithm

Our online learning rule operates through a unified mechanism where local Hebbian updates are accumulated in dual-timescale eligibility traces, then applied through fast and slow weight update pathways. This design addresses the core challenge of online adaptation: maintaining rapid responsiveness to immediate errors while preserving stable long-term learning.

**Notation.** For layer $\ell$, let $W^{(\ell)} \in \mathbb{R}^{n_{\mathrm{post}} \times n_{\mathrm{pre}}}$ be the weight matrix, $\mathrm{pre}_t^{(\ell)} \in \mathbb{R}^{n_{\mathrm{pre}}}$ the presynaptic activity at time $t$, $\mathbf{e}_t^{(\ell)} \in \mathbb{R}^{n_{\mathrm{post}}}$ a layer-local error drive, and $d_t^{(\ell)} \in \mathbb{R}^{n_{\mathrm{post}}}$ the postsynaptic sensitivity (LIF surrogate gradient; large near threshold). All vectors are column vectors; $\odot$ denotes elementwise product; outer products are written $a\, b^{\top}$. We set $\mathrm{pre}_t^{(1)} = \mathbf{x}_t$, $\mathrm{pre}_t^{(2)} = \mathbf{s}_t^{(1)}$, $\mathrm{pre}_t^{(3)} = \mathbf{s}_t^{(2)}$, and for the recurrent path use $\mathrm{pre}_t^{(\mathrm{rec})} = \mathbf{s}_{t-1}^{(1)}$.

**Three-Factor Hebbian Plasticity and Eligibility Traces.** *Layer-local error drives using current feedforward weights:*

$$\tilde{\mathbf{e}}_t^{(3)} := \mathcal{R}(\mathbf{y}_t - \hat{\mathbf{y}}_t), \quad \mathbf{e}_t^{(2)} := \left(W^{(3)}\right)^{\top} \tilde{\mathbf{e}}_t^{(3)}, \quad \mathbf{e}_t^{(1)} := \left(W^{(2)}\right)^{\top} \mathbf{e}_t^{(2)}.$$

The algorithm begins with three-factor Hebbian updates that capture coincident presynaptic activity, postsynaptic sensitivity, and error signals:

$$\Delta W_{\mathrm{hebb}}^{(\ell)}(t) = \left(\tilde{\mathbf{e}}_t^{(\ell)} \odot d_t^{(\ell)}\right) \left(\mathrm{pre}_t^{(\ell)}\right)^{\top}. \tag{1}$$

The same rule updates $W^{(\mathrm{rec})}$ with $\mathrm{pre}_t^{(\mathrm{rec})}$. Rather than applying these updates directly, each synapse accumulates them in dual eligibility traces with different decay timescales:

$$E_{\mathrm{fast}}^{(\ell)}(t) = \lambda_{\mathrm{fast}}\, E_{\mathrm{fast}}^{(\ell)}(t-1) + \Delta W_{\mathrm{hebb}}^{(\ell)}(t), \tag{2}$$

$$E_{\mathrm{slow}}^{(\ell)}(t) = \lambda_{\mathrm{slow}}\, E_{\mathrm{slow}}^{(\ell)}(t-1) + \Delta W_{\mathrm{hebb}}^{(\ell)}(t), \tag{3}$$

where $\lambda_{\mathrm{fast}} = \exp(-\Delta t/\tau_{\mathrm{fast}})$ and $\lambda_{\mathrm{slow}} = \exp(-\Delta t/\tau_{\mathrm{slow}})$. These are combined as:

$$E_{\mathrm{comb}}^{(\ell)}(t) = \alpha_{\mathrm{mix}}\, E_{\mathrm{fast}}^{(\ell)}(t) + \left(1 - \alpha_{\mathrm{mix}}\right) E_{\mathrm{slow}}^{(\ell)}(t). \tag{4}$$

Conceptually, these eligibility traces act as Hebbian accumulators: the fast trace emphasizes very recent three-factor updates, while the slow trace averages over a longer history. At each timestep,

the error-normalized term $\tilde{\mathbf{e}}_t^{(\ell)} \odot d_t^{(\ell)}$ is already folded into $\Delta W_{\text{hebb}}^{(\ell)}(t)$, so $E_{\text{fast}}^{(\ell)}$ and $E_{\text{slow}}^{(\ell)}$ store short- and long-term histories of reinforcement-modulated correlations at each synapse. Updates are temporally local (depending only on current activities and fixed-size traces), layer-local (each layer uses its own error drive and presynaptic activity), but spatially non-local in that the layer errors $\mathbf{e}_t^{(\ell)}$ are obtained by standard backpropagation across layers using current feedforward weights. In the intracortical BCI setting, this synapse-specific dual-timescale memory allows different outgoing targets of the same electrode to weight fast versus slow history differently, a degree of temporal expressiveness that neuron-level O($N$) factorization cannot provide and that our ablations in Table 6 empirically support.

**Dual-Timescale Weight Updates.** The eligibility traces drive two parallel update streams that address different temporal requirements (Pfister & Gerstner, 2006; Benna & Fusi, 2016). Fast updates apply the combined trace immediately,

$$W^{(\ell)} \leftarrow W^{(\ell)} + \eta_{\text{fast}}\, E_{\text{comb}}^{(\ell)}(t),$$

providing rapid adaptation. Simultaneously, a momentum-smoothed accumulator builds evidence for stable changes:

$$G^{(\ell)}(t) = \mu\, G^{(\ell)}(t-1) + (1-\mu)\, E_{\text{comb}}^{(\ell)}(t).$$

Every $K$ timesteps, this drives slow updates:

$$W^{(\ell)} \leftarrow W^{(\ell)} + \eta_{\text{slow}}\, \mathcal{R}\big(\bar{G}_K^{(\ell)}(t)\big),$$

where $\mathcal{R}(\cdot)$ provides RMS normalization and $\bar{G}_K^{(\ell)}(t)$ is the $K$-step average. This dual pathway balances plasticity with stability.

**Stability Mechanisms.** RMS-based normalisation keeps error and spike magnitudes within a stable range using exponential moving averages, and per-neuron weight projection caps the norm of each neuron's incoming weights to avoid runaway growth. The full algebraic forms and integer-friendly lookup-table implementations of these routines are given in Appendix D.

In summary, this learning rule combines dense per-timestep three-factor Hebbian updates, dual eligibility traces that separate fast adaptation from slower consolidation, and homeostatic normalisation to keep signals bounded. Ablation studies in Section 4 and closed-loop experiments show that together these components support continuous online adaptation without replay buffers or batched gradient updates.

**Complexity and Memory Analysis.**

Table 1: Peak training memory breakdown (MiB) for representative SNN architectures at $T = 120$ timesteps. Online SNN uses only static state; BPTT SNN adds a dynamic activation term that stores the time-unrolled computational graph and intermediate activations used by automatic differentiation, which scales with $T$.

| Component | Online 96-256-128-2 | BPTT 96-256-128-2 | Online 96-1024-512-2 | BPTT 96-1024-512-2 |
|---|---|---|---|---|
| Parameters $W+b$ | 0.47 | 0.47 | 6.38 | 6.38 |
| Eligibility traces $E_{\text{fast}}, E_{\text{slow}}$ | 0.94 | – | 12.76 | – |
| Momentum and RMS accumulators | 0.47 | – | 6.38 | – |
| Gradients | – | 0.47 | – | 6.38 |
| Optimizer state (Adam $m, v$) | – | 0.94 | – | 12.77 |
| Static total | 1.88 | 1.88 | 25.53 | 25.54 |
| Dynamic activations | 0.00 | 1.39 | 0.00 | 5.26 |
| **Peak total** | **1.88** | **3.27** | **25.53** | **30.80** |

**Time and memory.** As summarised in Table 1, the dynamic activations term for the BPTT SNN groups all tensors in the time-unrolled computational graph, including hidden states, membrane potentials, spike outputs, and intermediate quantities retained by PyTorch's automatic differentiation engine; the Online SNN has no such term and stores only static state. Per timestep, the online update requires current activations, integer RMS statistics, and two eligibility matrices per trained weight matrix, plus a momentum-like accumulator. Let $P$ denote the number of trainable parameters. The online SNN stores $(W + b)$ for parameters together with three additional $W$-sized buffers (fast traces, slow traces, and a momentum/RMS accumulator), yielding a static memory cost on the order of $4P$

that does *not* grow with sequence length $T$. In contrast, BPTT-based training stores the same order of static state (parameters, gradients, optimizer buffers) but in addition must retain $O(T)$ activations and intermediate quantities for backpropagation through time. Thus, training memory is $O(P)$ and *constant in $T$* for our method (parameters plus traces and statistics), whereas BPTT requires $O(P)$ static memory plus $O(T)$ activations for gradient computation. Both approaches have $O(P)$ spatial complexity and $O(TP)$ total compute, but the removal of the $O(T)$ activation term leads to the measured peak-memory reductions in Table 1 and becomes increasingly important for long sequences (Figure 8).

**Measured peak memory (MiB)** at 120 timestep sequences: the 96-256-128-2 architecture shows Online 1.88 vs BPTT 3.27, and the 96-1024-512-2 architecture shows Online 25.53 vs BPTT 30.80 as per Table 1. For longer sequences, Appendix G summarizes additional scaling experiments, which show that BPTT memory grows roughly linearly with $T$ while the Online SNN remains fixed, leading to progressively larger relative savings.

## 3 EXPERIMENTAL METHODS

**Datasets.** We evaluate on two public intracortical datasets: *MC Maze* (Churchland & Kaufman, 2022) with 10 ms resampling, an 80 ms kinematic lag, and 100 ms spike/velocity windows, and *Zenodo Indy* (O'Doherty et al., 2017) with 50 ms non-overlapping bins at zero lag. MC Maze consists of delayed centre-out reaches through fixed mazes, producing repeatable, trial-aligned trajectories; Zenodo Indy comprises self-paced reaches to dense target grids recorded over many sessions, yielding richer, less constrained kinematics. Targets are 2D velocities derived from positions. Our SNNs consume raw spike counts; Kalman and LSTM baselines consume rate-normalized inputs. Splits avoid leakage by using trial-wise splits on MC Maze and chronological splits on Zenodo.

**Baselines.** Kalman Filter, LSTM (Premchand et al., 2020), BPTT-SNN, and the proposed Online SNN share the same SNN architecture on each dataset, and we evaluate the Online SNN in batched and timestep-wise modes. All SNN variants use raw spike counts and baselines use normalized firing rates; baseline hyperparameters are given in Appendix J.0.4.

**Metrics.** We use Pearson correlation coefficient $R$ between predicted and true velocity components, reporting mean ± SEM across data splits. For closed-loop tasks, we measure time-to-target and use Welch's t-test for significance; full protocol details are in Appendix C.

## 4 RESULTS

We now evaluate the proposed online SNN in three stages: offline decoder comparison on intracortical datasets, component-wise ablations to understand which ingredients matter on which tasks, and closed-loop simulations to assess online supervised adaptation under non-stationarities and learning from scratch.

**Comparative Decoder Performance on Offline Datasets.**

We benchmarked the Online SNN (in its Batched Online mode for these offline tests) against a Kalman Filter (KF), a Long Short-Term Memory network (LSTM), and an offline-trained Spiking Neural Network using Backpropagation Through Time (BPTT-SNN). For the MC Maze dataset, the SNN architecture consisted of 182 input neurons (91 from M1 and 91 from PMd cortices) and followed a 182-1024-512-2 architecture; for the Zenodo Indy dataset, the architecture used 96 input neurons (from M1 cortex) and followed a 96-256-128-2 architecture, kept consistent between the Online SNN and BPTT-SNN for fair comparison. Performance, measured by Pearson correlation $R$ for X and Y velocity predictions, is shown in Figures 2a and 2b. Recurrent ANN baselines (LSTM in the main comparison, with additional GRU results summarized in Appendix K) attain the highest offline correlations, whereas the Online SNN remains slightly lower in $R$ but offers constant-in-$T$ training memory and true per-timestep online updates that the batch-trained BPTT-SNN and LSTM/GRU baselines do not provide.

**Ablation Studies**

Overall, the ablations indicate that three-factor gating, recurrence, RMS normalization, and dual-timescale eligibility traces all contribute to decoder robustness in a dataset-dependent way, with

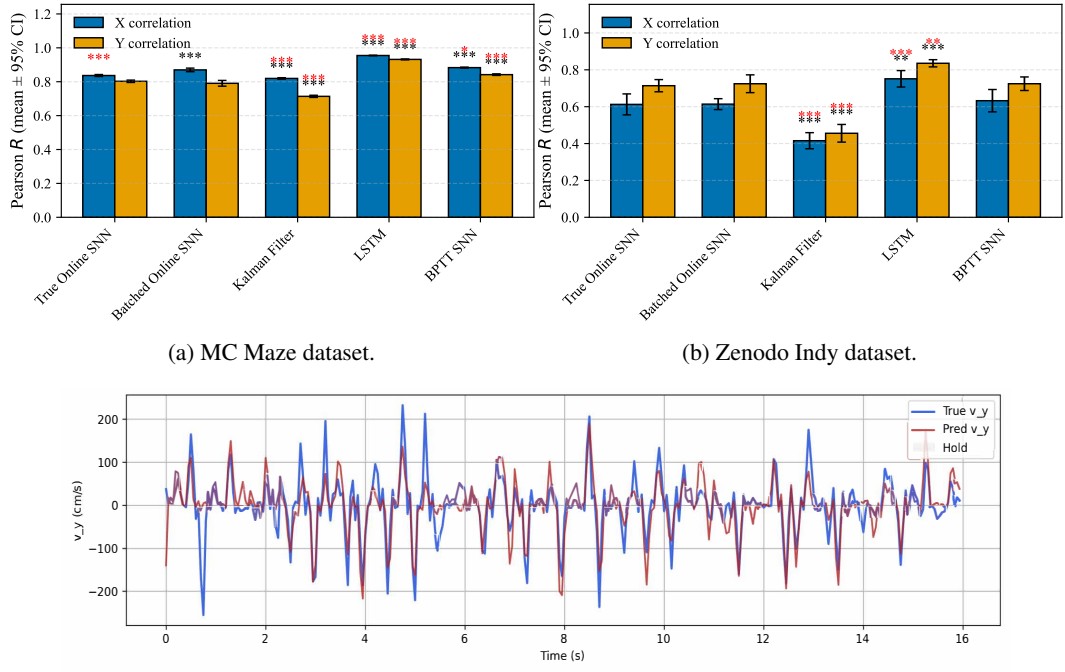

(a) MC Maze dataset.

(b) Zenodo Indy dataset.

(c) Sample of Zenodo Indy Y-velocity decoding.

Figure 2: Decoder performance: Pearson R, mean $\pm$ SEM, $n = 10$. **Order of Decoders in (a) and (b): True Online SNN, Batched Online SNN, Kalman Filter, LSTM, BPTT SNN.** MC Maze decoder comparison shown in (a), Zenodo Indy decoder comparison shown in (b). Performance of True (Timestep-wise) Online SNN on Zenodo Indy session 20161013_03 shown in (c). In figures (a) and (b), red significance stars indicate comparison with the Batched Online SNN; black significance stars indicate comparison with True (Timestep-wise) Online SNN. Together, these panels show that the Online SNN attains competitive correlations relative to the BPTT SNN and classical decoders while remaining below LSTM/GRU in offline accuracy.

Zenodo ablation trends visualised in Figure 3 and the full statistics reported in Appendix I and Appendix E. We additionally ablated the reward lookup table that scales the fast learning rate based on recent error magnitude. On Zenodo Indy in the Batched Online SNN regime (50 ms bins, 19 sessions), enabling the table increased mean test correlation from $R = 0.58 \pm 0.14$ to $R = 0.66 \pm 0.07$, with improvements on both axes (paired one-sided $t$-tests, $n = 19$: $p_x = 8.6 \times 10^{-3}$, $p_y = 5.3 \times 10^{-3}$, $p_{\text{avg}} = 2.6 \times 10^{-3}$ for the hypothesis $R_{\text{LUT}} > R_{\text{no-LUT}}$). In contrast, in the timestep-wise True Online SNN mode across 15 sessions, mean $R$ values for reward-lookup and constant-rate variants were similar and the same paired tests did not reject the null (all $p \approx 0.45$–$0.51$). This pattern suggests that reward-based scaling provides a statistically meaningful benefit when updates are accumulated over short Batched Online SNN sequences, where it sharpens plasticity during high-error epochs, while per-timestep RMS homeostasis already keeps errors well scaled in the true online regime so the additional lookup-based modulation offers limited further gain.

## 4.1 ONLINE SNN LEARNING DYNAMICS: EFFICIENCY AND ADAPTATION

A comparison of the Online SNN with the BPTT-SNN on the MC Maze dataset (Figure 4) shows that the Online SNN reaches strong validation correlations after only a few passes through the data, despite using sample-wise updates and an effective batch size of one. BPTT-SNN ultimately attains a higher final $R$ when trained for many more epochs, but requires substantially more optimizer steps and data exposure to reach that regime, highlighting the Online SNN's data efficiency in early training.

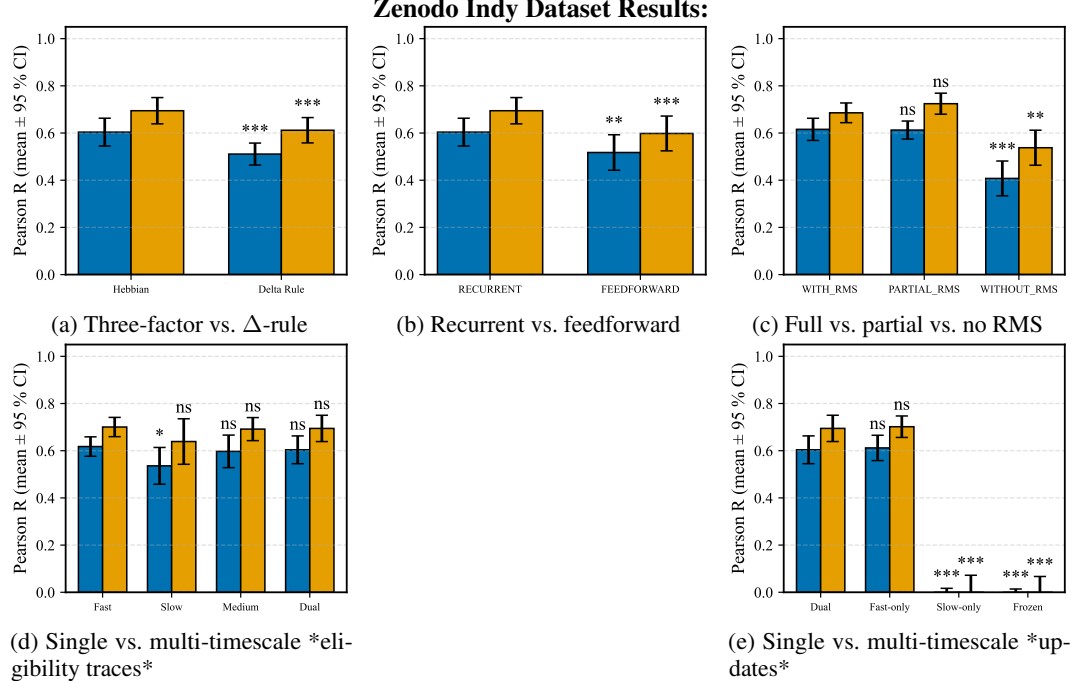

**Zenodo Indy Dataset Results:**

(a) Three-factor vs. Δ-rule

(b) Recurrent vs. feedforward

(c) Full vs. partial vs. no RMS

(d) Single vs. multi-timescale *eligibility traces*

(e) Single vs. multi-timescale *updates*

Figure 3: Zenodo Indy ablation study: update rules, recurrence, RMS, timescales, mechanisms. In (d), the order of timescale traces is Fast, Slow, Medium, Dual; In (e), the order of timescale updates is Dual, Fast-only, Slow-only, Frozen (i.e., no learning).

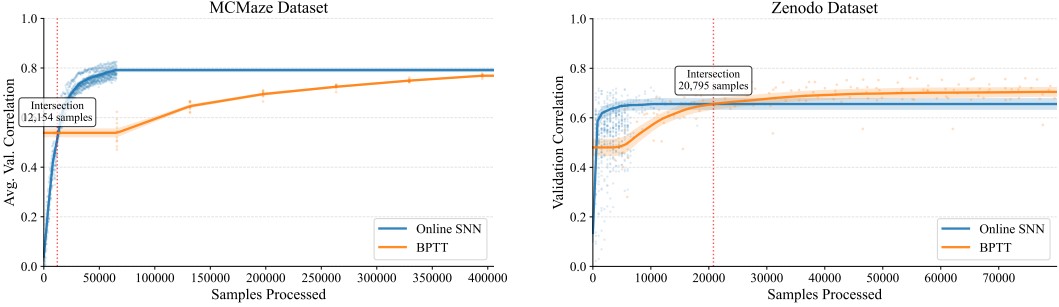

(a) MC Maze dataset learning curves averaged over 10 splits.

(b) Zenodo Indy dataset learning curves averaged over 10 sessions.

Figure 4: Learning curves: validation R vs. samples. Mean $\pm$ SEM, $n = 10$. Across both datasets, the Online SNN reaches strong validation correlations in fewer passes through the data than the BPTT SNN, which attains a higher final $R$ only after many more epochs.

## 4.2 CLOSED-LOOP SIMULATIONS WITH SYNTHETIC NEURAL POPULATIONS

**Disruption ablations (remapping, drift, dropout)** Decoder robustness to neural nonstationarities was assessed using three disruptions that mirror common intracortical failure modes: 90% remapping, where preferred directions are reassigned to mimic electrode shift or encapsulation; 90% drift, where firing rates are compressed and baseline activity is elevated; and 90% neuron dropout, where most recording sites are silenced (Figures 5a, 5b, 5c). Closed-loop hyperparameters are reported in Appendix J.0.5. Before disruption, across the first 150 reaches all decoders (Kalman Filter, LSTM, BPTT SNN, Online SNN) achieved similarly low mean time-to-target values below roughly 0.3 seconds, indicating that the synthetic task is well solved by a range of architectures when conditions are stable.

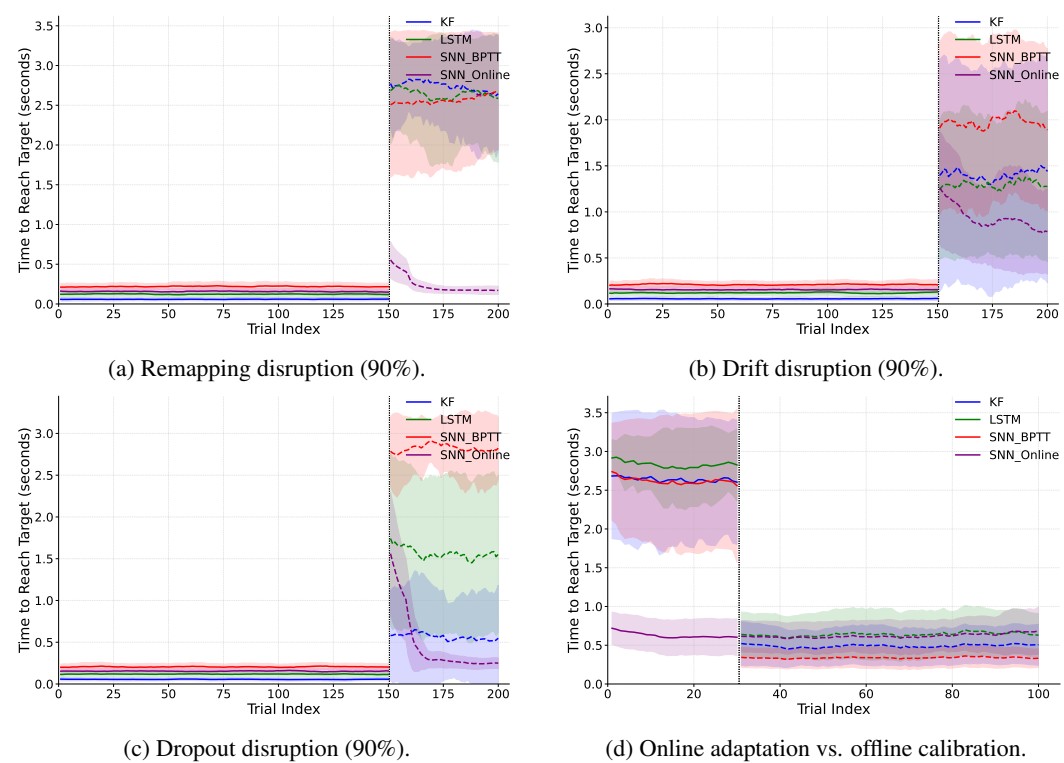

(a) Remapping disruption (90%).

(b) Drift disruption (90%).

(c) Dropout disruption (90%).

(d) Online adaptation vs. offline calibration.

Figure 5: Closed-loop adaptation to disruptions (a-c) and learning from scratch (d). Mean over 10 runs. Across all perturbations, the Online SNN rapidly readapts and returns to near pre-disruption reach times, whereas fixed-parameter decoders remain impaired unless explicitly recalibrated offline.

Once the disruption is introduced at trial 150, the trajectories separate. The Online SNN shows rapid readaptation under all three perturbations, with time-to-target initially increasing but returning to near pre-disruption levels within roughly 15–20 reaches: for remapping and dropout, mean reach times recover from peaks around 1.5 seconds back to at most about 0.3 seconds, and for drift they fall from about 1.5 seconds to roughly 0.75 seconds over a similar window. In contrast, the fixed-parameter decoders lose performance after disruption and do not recover over the same horizon: LSTM and BPTT SNN reach times remain above 1.5 seconds, while the Kalman filter degrades less under dropout but is still substantially slower than the Online SNN under remapping and drift. These patterns are consistent with the intended roles of the learning rule components: fast eligibility traces emphasise recent three-factor updates to correct for abrupt changes, slow traces and consolidation preserve useful pre-disruption structure, and RMS-based homeostasis keeps update magnitudes well scaled as firing statistics shift, together enabling continuous closed-loop adaptation.

**Online adaptation versus offline calibration (no-pretrain comparison)** To contrast continuous online learning with conventional calibration pipelines, a second protocol (Figure 5d) compared decoders initialised randomly and given access only to data they generate themselves. During the first block of about 30 reaches, only the Online SNN updates its weights on every timestep, improving from reach times around 0.75 seconds to approximately 0.6 seconds, whereas the fixed-weight decoders exhibit highly variable behaviour with frequent timeouts at the 3-second limit.

In a second block of roughly 70 reaches, the offline-trainable decoders are recalibrated on their own collected data and then reinserted into the loop; after this batch calibration they achieve control quality similar to the already-adapted Online SNN, which maintains its earlier performance. This comparison highlights a complementary trade-off: offline-trained decoders can perform well once sufficient interaction data have been gathered and processed, but they provide little utility during the data-collection phase itself, whereas the Online SNN delivers usable closed-loop performance from random initialisation and continues to adapt as conditions evolve.

## 5 CONCLUSION

Conceptually, our contribution is to reinterpret eligibility traces for online SNN learning in intracortical BCI as dual-timescale Hebbian accumulators of completed three-factor updates rather than as gradient surrogates or error-bridging traces. This representation allows a fully online, optimizer-free learning rule in which each synapse maintains fast and slow reinforcement-modulated correlation histories that support rapid correction and longer-term consolidation while keeping training memory constant in sequence length. Within this framework, recurrence, RMS homeostasis, and synapse-specific traces are design choices targeted at noisy, heterogeneous Utah-array recordings (Maynard et al., 1997; Churchland & Shenoy, 2007; La Camera et al., 2006; Woeppel et al., 2021; Forrest et al., 2024; Patel et al., 2023) and diverse cortical neuron dynamics (Luebke & Chang, 2007; Prince et al., 2021; Wilbers et al., 2023), and empirically the factorization ablation in Table 6 suggests that synapse-level temporal expressiveness is beneficial when different downstream targets of the same electrode require different fast/slow mixtures.

Our three-factor rule differs from classic reward-modulated STDP in that it uses a dense kinematic error for frame-wise credit assignment while remaining local in time at the synapse. Surrogate gradients provide a sensitivity gate that confines plasticity to neurons near threshold, and the learning rule avoids the temporal weight-transport and memory costs of BPTT by not unrolling the network in time (Seung, 2003; Izhikevich, 2007; Frémaux & Gerstner, 2016; Neftci et al., 2019; Bellec et al., 2020; Zenke & Vogels, 2021). At the same time, the method is *spatially* non-local because hidden-layer error signals are obtained via standard backpropagation across layers, so we view it as *partially* biologically plausible: it incorporates spiking dynamics and three-factor Hebbian plasticity with dual timescales, but does not yet resolve the spatial weight-transport problem (Lillicrap et al., 2020).

Within a broader neurobiological perspective, the dense error signal $\mathbf{y}_t - \hat{\mathbf{y}}_t$ that drives our three-factor term can be interpreted as a prediction error broadcast rather than a literal backpropagated gradient: in cortex–basal-ganglia loops, neuromodulatory systems convey reward and performance errors that modulate local Hebbian plasticity (Frémaux & Gerstner, 2016; Seung, 2003; Izhikevich, 2007). In the BCI setting, this suggests viewing the decoder's error as an engineered analogue of such modulatory signals, providing task feedback to the synapses without claiming a direct one-to-one mapping onto specific biological pathways.

Integrating fast/slow consolidation directly into the online updater resolves a stability–plasticity tension without replay or mini-batches: fast traces enable rapid correction and slow consolidation preserves durable structure, echoing early/late LTP accounts and synaptic memory theories (Fusi et al., 2005; Benna & Fusi, 2016; Zenke et al., 2017; Kirkpatrick et al., 2017). From a systems perspective, this dual-timescale Hebbian accumulator architecture provides constant-in-$T$ training memory while maintaining $O(P)$ spatial complexity, enabling continuous adaptation without storing temporal activation graphs. Practically, closed-loop simulations show that the updater maintains control under remapping, drift, and dropout and can learn from scratch, supporting long-running BCIs in non-stationary conditions when supervised targets or task-related feedback are available.

Relative to other temporally local rules such as OTTT, S-TLLR, SLTT, and TESS (Xiao et al., 2022; Apolinario & Roy, 2025; Meng et al., 2023; Apolinario et al., 2025), which factorize eligibility into neuron-level $O(N)$ traces or use fixed feedback bases and are tuned on large-scale computer vision data with floating-point optimizers, our design keeps synapse-specific dual-timescale Hebbian accumulators with integer-friendly RMS homeostasis targeted at Utah-array-scale intracortical BCI. Direct empirical comparison on shared benchmarks is left to future work, but our factorization ablations and memory analysis suggest that this $O(N^2)$ design remains practical at typical BCI scales while providing better accuracy under heterogeneous spiking inputs. Overall, we present a BPTT-free online SNN decoder that combines temporally local three-factor plasticity with dual-timescale consolidation and integer-friendly RMS homeostasis to achieve competitive decoding relative to BPTT-trained SNNs with training memory that is constant in sequence length and continuous adaptation to non-stationarities, while still relying on spatial backpropagation across layers and trading some offline accuracy compared to LSTM/GRU decoders; key limitations are summarized in Appendix A.

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

## A  LIMITATIONS

Closed-loop results use synthetic neural populations; validation on chronic human recordings is still required  (Cunningham et al., 2011). The consolidation window $K$, LIF decay parameters, and reset thresholds are hand-tuned, and fully automatic scheduling would improve robustness across tasks. Empirical scaling on neuromorphic devices also remains to be demonstrated, despite favorable memory analysis and specific engineering optimization choices. All training and memory measurements in this work are obtained from PyTorch/snnTorch (Eshraghian et al., 2023) implementations on conventional GPUs using standard floating-point arithmetic; porting the integer-friendly design to fixed-point or mixed-precision neuromorphic hardware and repeating these measurements is future work. Conceptually, the learning rule is only partially biologically plausible because hidden-layer errors are still delivered by spatially non-local backpropagation; integrating the dual-timescale Hebbian accumulator framework with more local feedback mechanisms, such as feedback alignment or basis-feedback schemes, is an important direction for future work. Finally, we restrict empirical comparisons to Kalman, MLP, LSTM, GRU, and BPTT-SNN baselines that are standard for intracortical BCI; recent local rules such as OTTT, S-TLLR, SLTT, and TESS were developed and tuned for homogeneous computer-vision inputs with factorised $O(N)$ traces, whereas our own neuron-level factorisation ablations indicate that such designs underperform synapse-specific traces on heterogeneous Utah-array recordings 6, so adapting and fairly tuning those methods, and the ilk of transformer-style decoders and TCNs to this intracortical setting is left to future work. We also focus on online supervised adaptation where target kinematics or task-related feedback are available, leaving unsupervised or self-supervised extensions of the framework to future work.

## B  NOTATION

Table 2: Key notation and symbols used in the learning rule and architecture.

| Symbol | Description |
|---|---|
| $\mathbf{x}_t$ | Input spike count vector at time $t$ |
| $\mathbf{y}_t$ | Target 2D velocity at time $t$ |
| $\hat{\mathbf{y}}_t$ | Predicted 2D velocity at time $t$ |
| $\mathbf{u}_t^{(k)}$ | Membrane potentials for layer $k$ at time $t$ |
| $\mathbf{s}_t^{(k)}$ | Spike outputs for layer $k$ at time $t$ |
| $d_{\text{LIF}}$ | Surrogate gradient (neuronal sensitivity) |
| $E^{\text{fast}}, E^{\text{slow}}$ | Fast and slow eligibility traces |
| $\lambda_{\text{fast}}, \lambda_{\text{slow}}$ | Decay coefficients: $e^{-\Delta t/\tau_{\text{fast/slow}}}$ |
| $\tau_{\text{fast}}, \tau_{\text{slow}}$ | Time constants for eligibility traces (120ms, 700ms) |
| $\alpha_{\text{mix}}$ | Mixing coefficient for combining fast/slow traces |
| $\eta_{\text{fast}}, \eta_{\text{slow}}$ | Fast and slow learning rates |
| $K$ | Consolidation window (timesteps) |
| $\mathcal{R}(\cdot)$ | RMS normalization function |

## LLM USAGE DISCLOSURE

Large language models (LLMs) were used in this work in following ways. Draft text for all sections, figure captions, and code for experiments/analysis were initially generated with LLM assistance and then edited, verified, and curated by the authors. The authors are solely responsible for the accuracy of all statements, correctness of code, and validity of results.

**Human verification & responsibility.** All claims, equations, proofs/derivations, and references were checked by the authors; all experiments were re-run by the authors from clean environments; and all plots/tables were regenerated from verified outputs.

**Confidentiality & ethics.** No confidential third-party material (e.g., peer reviews, private manuscripts) was provided to any LLM. We did not include hidden prompt-injection text in the submission. All

external data/code obeys their licenses, and references were verified to correspond to real, relevant sources.

This disclosure is mirrored in the submission form as required by ICLR policy.

## C  EXPERIMENTAL METHODS

We evaluate the Online SNN decoder as follows. Before assessing the continuous online adaptation capabilities of the proposed Online SNN in simulated closed-loop scenarios, we first establish its fundamental decoding efficacy in a more controlled, offline setting. This offline evaluation allows comparison against established batch-trained decoders using standardized datasets and metrics, validates the core learning mechanisms by demonstrating that the model can learn complex neural-to-behavioral mappings from static datasets, and provides a baseline performance characterization before introducing the complexities of closed-loop interaction and non-stationarities. The experimental protocol—encompassing data splitting, preprocessing, training procedures, and evaluation metrics—is kept consistent across all models to ensure fair and unbiased comparisons.

### C.1  DATASETS AND PREPROCESSING

To ensure robust, generalizable findings, we evaluate the decoders on two distinct, publicly available non-human primate intracortical recording datasets (Churchland & Kaufman, 2022; O'Doherty et al., 2017). Both datasets are made available under permissive licenses: the MC Maze dataset is available through DANDI under CC-BY-4.0, and the Zenodo Indy dataset is available under CC-BY-4.0 license. The choice of these datasets is motivated by their widespread use in the BCI research community, the richness of the neural data they provide, and the different behavioral tasks they represent, thereby testing the adaptability of our model to varied neural encoding schemes.

**MC Maze Dataset**   This dataset, from the Neural Latents Challenge 2023, contains intracortical multi-unit recordings from a macaque performing a 2D maze navigation task. Its inclusion is valuable due to its recency and the complexity of the continuous control task. Data are accessed using standard neurodata format utilities. Raw neural and kinematic data are initially resampled to a 10 ms resolution. For each trial, data are aligned to movement onset; kinematics are lagged by 80 ms relative to spikes, following the dataset authors' recommendation for best decode performance. Within each aligned trial, continuous data streams (spikes and hand velocity) are binned into 100 ms windows with a 10 ms stride. The spike input for each 100 ms window is represented by the raw summed spike count from each neuron. True hand-velocity targets $(v_x, v_y)$ are derived by numerically differentiating the cursor position data and then averaging the velocity values within each 100 ms window. These target velocities are z-score normalized using statistics derived solely from the training set to prevent data leakage and ensure that the model learns from a standardized output distribution. For the Online SNN evaluation, spike inputs remain as raw counts and are unnormalized. For the Kalman Filter and LSTM models, spike counts are converted to rates (counts per second by dividing by the 100 ms bin width) and then z-score normalized using training set statistics. Data splitting is performed on entire trials (70%/15%/15% for train/validation/test) to prevent temporal correlations from leaking between sets, ensuring a more rigorous evaluation of generalization.

**Zenodo Indy Reaching Dataset**   This dataset comprises multi-unit spiking activity from a primate performing center-out reaching movements, a classic BCI paradigm. Its inclusion allows comparison with a large body of existing literature. Raw spike times are loaded directly from data files. Spike events are binned into 50 ms non-overlapping windows, with each window encoded by the raw spike counts from each neuron. To derive cursor velocity, the raw cursor position data is first smoothed using a 4th-order, 10Hz low-pass Butterworth filter, consistent with standard practices for this dataset. The filtered position is then numerically differentiated, and the resulting velocity is mean-averaged within each 50 ms window. A zero-millisecond kinematic lag is used, meaning neural activity and cursor velocity are aligned to the same time bin. Target velocities are z-score normalized using training set statistics. For the Online SNN evaluation, spike features remain as raw counts and are unnormalized. For the Kalman Filter and LSTM models, spike counts are converted to rates (counts per second) and then z-score normalized using training set statistics. Data are split chronologically: the initial 70% of the session's data is used for training, the subsequent 15% for validation, and the

final 15% for testing. This chronological split better reflects a real-world scenario where a decoder is trained on past data and tested on future, unseen data within the same recording session.

To assess sensitivity to bin width, we evaluated seven window sizes between 32 ms and 256 ms on five randomly selected Zenodo sessions. Performance was highest for bins around 30–50 ms and degraded substantially for very coarse bins (200–256 ms), consistent with the loss of behaviorally relevant timing information when excessive temporal averaging is applied. Table 3 summarizes the mean test correlations across this sweep, supporting our choice of 50 ms as a robust trade-off between spike count reliability and temporal resolution.

Table 3: Zenodo Indy bin-width sensitivity: mean test correlations (mean $\pm$ SEM) across five sessions for X and Y velocity components.

| Bin width (s) | Mean corr (X) | Mean corr (Y) |
|:---:|:---:|:---:|
| 0.032 | $0.6365 \pm 0.1264$ | $0.7635 \pm 0.0514$ |
| 0.050 | $0.6105 \pm 0.1066$ | $0.7271 \pm 0.0898$ |
| 0.064 | $0.5683 \pm 0.1332$ | $0.6709 \pm 0.1444$ |
| 0.100 | $0.3982 \pm 0.1666$ | $0.5441 \pm 0.1983$ |
| 0.128 | $0.4236 \pm 0.1345$ | $0.4117 \pm 0.1189$ |
| 0.200 | $0.1906 \pm 0.0664$ | $0.3199 \pm 0.2104$ |
| 0.256 | $-0.0175 \pm 0.1738$ | $0.1542 \pm 0.1714$ |

## C.2 DECODER ARCHITECTURES AND TRAINING PROCEDURES

The **Online SNN** can be evaluated in two distinct modes:

**Batched Online SNN:** For offline evaluation on static datasets, this mode processes data in fixed-length, overlapping sequences (10 timesteps with 50% overlap, creating sequences at timesteps 0-9, 5-14, 10-19, etc.) grouped into mini-batches. The overlapping design ensures temporal continuity while enabling efficient batch processing, and the 10-timestep length provides sufficient context for short-term temporal dependencies without excessive computational overhead.

**Timestep-wise Online SNN:** For analyzing real-time adaptation, this mode processes data one timestep at a time, with hidden states persisting chronologically to mimic continuous BCI operation. This directly simulates a live, sample-by-sample decoding scenario. State resets can be applied at logical breaks (e.g., trial boundaries). This mode of the Online SNN can be trained for a fewer number of epochs due to the per-timestep updates.

This distinction helps understand the model's behavior in different deployment contexts. Our hypothesis is that this online, local, and biologically-inspired learning mechanism can achieve competitive performance. It offers advantages in adaptability and computational efficiency.

All neural network decoders (LSTM, BPTT SNN, Online SNN) are implemented in PyTorch and `snnTorch` (Eshraghian et al., 2023). Training and evaluation are conducted using a consistent data processing pipeline.

We evaluate the Online SNN's adaptive capabilities through closed-loop BCI simulations using synthetically generated neural data. Details of the simulation methodology are provided in Appendix J.

Reproducibility and compute details are summarized here: complete offline hyperparameters are given in Appendix J.0.4, closed-loop hyperparameters in Appendix J.0.5, and all experiments were re-run from clean environments on PyTorch/snnTorch implementations using a single A100 or H100 GPU, with each configuration evaluated over 10 independent runs and Welch's t-tests used for significance.

# D  Learning Algorithm Details

This section expands the dual-timescale Hebbian accumulator learning rule introduced in Section 2.2, emphasising that the eligibility traces integrate completed three-factor updates rather than approximate backpropagated gradients.

---

**Algorithm 1:** Per-Timestep Online Update
**Input:** Spikes $\mathbf{x}_t$, target velocity $\mathbf{y}_t$
**State:** Weights $W$, membrane potentials $\mathbf{u}(t)$, traces $E^{\text{fast}}, E^{\text{slow}}$

---

1. **Forward pass:** $\mathbf{u}(t) \leftarrow \beta \mathbf{u}(t-1) + W\mathbf{x}(t)$; get $\hat{\mathbf{y}}_t, d_{\text{LIF}}$
2. **Error computation:** $\mathbf{e}(t) \leftarrow \mathbf{y}(t) - \hat{\mathbf{y}}(t)$
3. **RMS normalization:** Update integer EMAs; $\tilde{\mathbf{e}}(t) \leftarrow \mathbf{e}(t) \cdot \text{LUT}[\text{RMS}]$
4. **Three-factor term:** $\Delta W^{\text{hebb}}(t) \leftarrow (\tilde{\mathbf{e}}(t) \odot d_{\text{LIF}}(t))(\text{pre}_t)^T$
5. **Update traces:** $E^{\text{fast}} \leftarrow \lambda_f E^{\text{fast}} + \Delta W^{\text{hebb}}(t)$
6. $\qquad\qquad E^{\text{slow}} \leftarrow \lambda_s E^{\text{slow}} + \Delta W^{\text{hebb}}(t)$
7. **Combined trace:** $E^{\text{comb}} \leftarrow \alpha_{\text{mix}} E^{\text{fast}} + (1 - \alpha_{\text{mix}}) E^{\text{slow}}$
8. **Fast update:** $W \leftarrow W + \eta_{\text{fast}} E^{\text{comb}}$
9. **Accumulate:** $G \leftarrow \mu G + (1-\mu) E^{\text{comb}}$
10. **if** $t \bmod K = 0$: $W \leftarrow W + \eta_{\text{slow}} \cdot \mathcal{R}(G)$
11. $\qquad\qquad$ Apply weight decay

---

**Memory:** $O(P)$ static, constant in sequence length $T$ (parameters + 2 traces + statistics)
**Compute:** $O(P)$ per timestep, no backprop graph storage

Figure 6: Complete per-timestep update procedure showing constant-in-$T$ training memory scaling.

This appendix provides the complete derivations and implementation details for the three-component learning system introduced in Section 2.2.

## D.1  Eligibility Traces

To enable credit assignment over recent activity and to bridge the gap between instantaneous three-factor updates and long-term consolidation, we maintain two sets of eligibility traces for each synaptic weight matrix: a fast trace ($E^{\text{fast}}(t)$) and a slow trace ($E^{\text{slow}}(t)$). These traces accumulate the instantaneous three-factor updates and decay exponentially over time, reflecting a form of working memory for synaptic plasticity (Bellec et al., 2020; Zenke & Ganguli, 2018; Frémaux & Gerstner, 2016). While related to the traces in e-prop, which are derived to approximate BPTT gradients, our traces are implemented as direct, hardware-friendly exponential moving averages of three-factor products rather than as surrogates for a gradient computation. Conceptually, the evolution of an eligibility trace $E$ is governed by a first-order differential equation:

$$\tau_e \frac{dE}{dt} = -E + \Delta W^{\text{hebb}}$$

where $\tau_e$ is the trace time constant and $\Delta W^{\text{hebb}}$ is the instantaneous three-factor update. The discrete update rules, applied at each time step $t$, are:

$$E^{\text{fast}}(t) = E^{\text{fast}}(t - \Delta t) \cdot e^{-\Delta t/\tau_{\text{fast}}} + \Delta W^{\text{hebb}}(t) \tag{2}$$

$$E^{\text{slow}}(t) = E^{\text{slow}}(t - \Delta t) \cdot e^{-\Delta t/\tau_{\text{slow}}} + \Delta W^{\text{hebb}}(t) \tag{3}$$

where $\tau_{\text{fast}}$ (e.g., 120 ms) and $\tau_{\text{slow}}$ (e.g., 700 ms) are the time constants for the fast and slow traces respectively. The decay coefficients, $e^{-\Delta t/\tau}$, are derived directly from these time constants. (In online mode, these time constants are scaled: $\tau_{\text{fast}}$ is halved and $\tau_{\text{slow}}$ is scaled by 0.8 for faster adaptation.) These two timescales allow the system to maintain sensitivity to recent, rapid changes while also preserving a longer-term memory of relevant activity. A combined eligibility trace, $E^{\text{comb}}(t)$, is then formed as a linear mixture of the fast and slow traces:

$$E^{\text{comb}}(t) = \alpha_{\text{mix}} \cdot E^{\text{fast}}(t) + (1 - \alpha_{\text{mix}}) \cdot E^{\text{slow}}(t) \tag{4}$$

where $\alpha_{\text{mix}}$ is a mixing coefficient (e.g., 0.5), which determines the relative influence of the fast and slow eligibility traces. This combined trace is what drives the effective fast weight updates applied to the network at each timestep:

$$\Delta W(t) = \eta_{\text{fast}} \cdot E^{\text{comb}}(t),$$

where $\eta_{\text{fast}}$ is the fast learning rate that scales immediate, per-timestep plastic changes. It controls the responsiveness of the working weights to the most recent activity and error.

## D.2 TWO-TIMESCALE WEIGHT CONSOLIDATION

Online learning systems must balance plasticity and stability, especially in non-stationary environments like BCIs: the system must be plastic enough to adapt to new information or changes in input statistics, yet stable enough to retain previously learned knowledge. To address this, our model incorporates a two-timescale weight consolidation mechanism. This approach is inspired by biological theories of memory consolidation, where learning initially induces labile synaptic changes (fast timescale) that are later consolidated into more enduring forms (slow timescale) (Fusi et al., 2005; Benna & Fusi, 2016; Kirkpatrick et al., 2017; Zenke et al., 2017). By maintaining separate fast-changing (transient) and slow-changing (stable) weight components, our system can rapidly respond to immediate errors via the fast Hebbian updates, while gradually integrating persistent changes into the more stable slow weights. This prevents catastrophic forgetting and promotes long-term learning. The combined eligibility trace $E^{\text{comb}}(t)$ is accumulated into a momentum-filtered accumulator $G(t)$, which smooths out noisy single-timestep updates:

$$G(t) = \mu G(t-1) + (1-\mu)E^{\text{comb}}(t), \tag{5}$$

where $\mu$ is the momentum coefficient (e.g., 0.9). Periodically (every $K$ timesteps, where $K$ is a hyperparameter balancing responsiveness and stability), these accumulated, smoothed changes (represented by $G(t)$) are averaged over the $K$ timesteps, then normalized by their Root Mean Square (RMS) value using the function $\mathcal{R}(\cdot)$, and finally consolidated into the slow weights, which are the weights actually used for inference:

$$W^{\text{slow}}(t+K) = (1-\alpha) W^{\text{slow}}(t) + \alpha \, \mathcal{R}(\bar{G}_K(t)), \tag{6}$$

where $\alpha$ is the slow learning rate, and $\bar{G}_K(t)$ denotes the average of $G(s)$ over the preceding $K$ timesteps. The function $\mathcal{R}(X) = X/\left(\sqrt{\text{mean}(X^2)} + \epsilon\right)$ normalizes the gradient by its RMS, ensuring stable updates regardless of gradient magnitude. In this two-rate scheme, $\eta_{\text{fast}}$ scales instantaneous updates applied continuously through the eligibility pathway, whereas $\alpha$ (equivalently, a slow step size $\eta_{\text{slow}}$) scales the periodic consolidation that integrates smoothed evidence into the stable weights every $K$ timesteps. This separation allows rapid, potentially transient adaptations without immediately overwriting the stable substrate, preserving past knowledge while still permitting continuous improvement.

To prevent overfitting, during each slow (consolidative) update, we apply a 1e-5 decay on the weight matrices.

## D.3 STABILITY AND HOMEOSTASIS

Continuous online learning, especially with three-factor rules that reinforce correlated activity, can be prone to instabilities such as runaway synaptic weights or saturated neuronal activity. To ensure stable long-term operation, which is necessary for any practical BCI system, our model employs two biologically-inspired homeostatic mechanisms. Homeostatic plasticity refers to the set of processes that neurons and neural circuits use to maintain their electrical activity within a stable physiological range, despite perturbations or ongoing synaptic modifications (Abbott & Nelson, 2000; Turrigiano & Nelson, 2004). These mechanisms are vital for preventing pathological states like epilepsy (runaway excitation) or silent networks (insufficient activity) and are implemented in our model as follows:

1. **Per-Neuron Weight Scaling and Clipping:** After each timestep, the incoming synaptic weights for each individual neuron are scaled and clipped to prevent weight explosion. This mechanism calculates the L2 norm of each row (representing incoming weights to a neuron) and rescales weights that exceed a predefined maximum threshold through power-of-two division operations.

This prevents any single neuron's weights from exploding, thereby maintaining stable total synaptic drive and preventing numerical instability. The power-of-two scaling approach is chosen for hardware efficiency, as it can be implemented using simple bitwise operations on neuromorphic processors.

2. **Root Mean Square (RMS)-based Normalization of Error and Spike Activities with Reward Lookup Modulation:** Intermediate error signals and pre-synaptic spike activities are dynamically normalized using RMS exponential moving averages, implemented using a 256-entry lookup table that maps RMS values to their inverse square roots for hardware-efficient computation (Davies et al., 2018; Hasssan et al., 2024). On top of this, a small 16-entry reward lookup table maps the normalized output-error magnitude into a discrete gain that scales the fast update, so larger recent errors transiently increase plasticity while preserving an integer-friendly implementation. Together, these mechanisms keep signal magnitudes and effective learning rates well scaled across varying neural activity levels and non-stationarities without relying on floating-point adaptive optimizers (Neftci et al., 2019).

These mechanisms ensure system resilience and stable performance during continuous adaptation with minimal computational overhead.

## E    ABLATION STUDIES RESULTS

Three-factor and delta rules differ in how synaptic change is gated. The delta rule scales the presynaptic activity by a postsynaptic error term (supervised, stable, but agnostic to a neuron's instantaneous spiking sensitivity). Our three-factor rule further gates by $d_{\text{LIF}} = \partial S/\partial u$, which is large only when a neuron's membrane potential is near threshold, concentrating plasticity on units poised to change their spiking. This distinction is minor on MC Maze but decisive on Zenodo Indy because the datasets differ in temporal alignment, task structure, and noise:

Table 4: Summary of key design components and their contribution relative to a simplified $\delta$-rule baseline on Zenodo and MC Maze. Values are percentage gains in average correlation.

| Innovation | Design feature | Zenodo gain vs. $\delta$-rule | MC Maze gain vs. $\delta$-rule |
|---|---|---|---|
| **RMS normalization** | Bitshift RMS EMAs (online $T = 1$ homeostasis) | +27.2% vs. without RMS | +0.8% vs. without RMS |
| **Recurrence** | Recurrent layer ($256 \rightarrow 256$) | +14.1% vs. feedforward | +0.7% vs. feedforward |
| **Hebbian accumulation** | $E = \sum_t \lambda^t \cdot \text{pre} \cdot \text{post} \cdot \text{error}$ | +13.5% vs. $\delta$-rule | −0.1% vs. $\delta$-rule |

Table 5: Effect of dual-timescale eligibility traces on robustness across Zenodo and MC Maze tasks. A single fast or slow trace is optimal for one dataset but suboptimal for the other; the dual configuration provides near-optimal performance on both without retuning.

| Timescale configuration | Zenodo corr | MC Maze corr | Effect |
|---|---|---|---|
| $\tau_{\text{fast}}$ only (120 ms) | 0.659 | 0.724 | Best on Zenodo, −5.7% vs. $\tau_{\text{slow}}$ on MC Maze |
| $\tau_{\text{slow}}$ only (700 ms) | 0.588 | 0.782 | Best on MC Maze, −9.5% vs. $\tau_{\text{fast}}$ on Zenodo |
| **Dual** ($\tau_{\text{fast}}, \tau_{\text{slow}}$) | **0.649** | **0.767** | Near-optimal on both, fixed hyperparameters |

In the RMS ablations, the three conditions correspond to full RMS (normalization of output errors, hidden-layer errors, and spike activities using running root-mean-square statistics), partial RMS (normalization applied only to the output error and spike activities, with hidden-layer errors left unnormalized), and no RMS (all normalization disabled).

1) Temporal alignment and lag structure. For MC Maze (Figure 7), decoding is performed with an 80 ms kinematic lag relative to spikes, which increases signal-to-noise by aligning neural drive to ensuing velocity. This alignment reduces the need for fine-grained sensitivity gating: error gradients already correlate well with the presynaptic drive, so delta updates approximate the gated rule and performance is nearly identical. Zenodo Indy provides continuous, self-paced reaches without explicit trial alignment or standardized neuro-to-kinematic lag. Error signals are temporally broader and less synchronized with presynaptic events. Here $d_{\text{LIF}}$ gates learning to moments when postsynaptic neurons are near threshold, sharpening credit assignment and yielding large gains ($\Delta R \approx 0.1$–0.2).

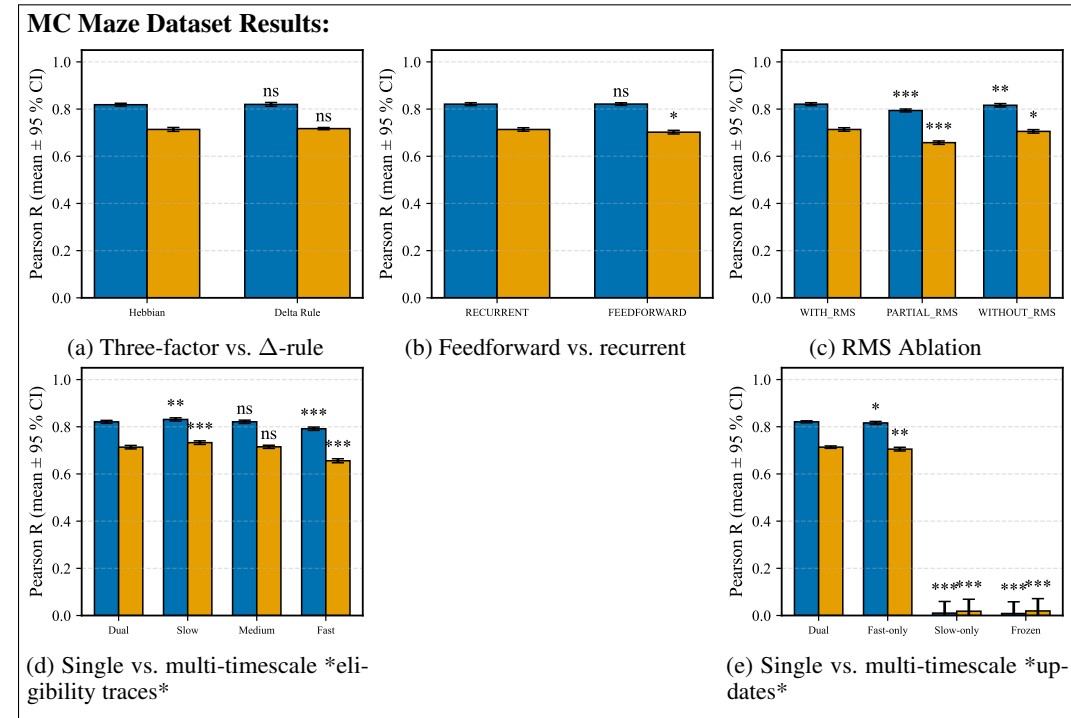

**MC Maze Dataset Results:**

(a) Three-factor vs. Δ-rule

(b) Feedforward vs. recurrent

(c) RMS Ablation

(d) Single vs. multi-timescale *eligibility traces*

(e) Single vs. multi-timescale *updates*

Figure 7: MC Maze ablation study: update rules, recurrence, RMS, timescales, mechanisms. In (d), the order of timescale traces is Fast, Slow, Medium, Dual; In (e), the order of timescale updates is Dual, Fast-only, Slow-only, Frozen (i.e., no learning).

2) Task and temporal statistics. MC Maze movements are constrained by trial structure and alignment to move onset, producing stereotyped dynamics and smoother velocity trajectories within windows, which makes coarse error-driven updates sufficient. Zenodo Indy includes self-paced reaches to dense target grids, richer kinematic variability, and session-to-session heterogeneity. Sensitivity gating prioritizes updates during informative transients and suppresses updates during quiescent or saturated regimes, improving bias–variance trade-offs under these conditions.

3) Noise and spiking sparsity. Zenodo Indy sessions often include higher unit counts and mixed sorting quality (including unsorted threshold crossings), leading to sparser and noisier effective inputs. The three-factor gate down-weights updates when postsynaptic units are far from threshold (low $d_{\mathrm{LIF}}$), mitigating noise amplification that the delta rule would otherwise introduce. MC Maze's binning plus lag yields higher effective SNR, so the delta rule is already close to optimal.

Empirically, the ablations reflect these factors: on Zenodo Indy, replacing three-factor with delta degrades both axes with statistically significant drops, whereas on MC Maze the difference is negligible. In short, $d_{\mathrm{LIF}}$ confers robustness when timing is less controlled, dynamics are more varied, and spike signals are noisier; when alignment and structure improve effective SNR (as in MC Maze with an 80 ms lag), delta updates suffice.

For MC Maze, the ablations indicate that the three-factor and delta rules perform nearly identically, with differences that are very small and not practically meaningful. Architectural recurrence confers a consistent, modest gain concentrated on Y while X remains similar across architectures, suggesting a limited but reliable contribution from temporal memory. Full RMS normalization provides the strongest results, no normalization is close behind, and partial normalization is noticeably worse; consequently, partial normalization should be avoided. With regard to trace timescales, slow traces are strongest, medium and dual traces are close, and fast traces are weakest, indicating that longer temporal integration is beneficial on this dataset. In terms of update mechanisms, using both fast and slow updates performs best, fast-only is slightly lower, and slow-only or frozen updates collapse performance toward zero, underscoring the necessity of rapid adaptation. Taken together, these ablations show that on Zenodo Indy the three-factor gate, RMS homeostasis, recurrence, and dual-

timescale traces are all important for robustness in a noisier, less aligned regime, whereas on MC Maze performance is dominated by recurrence and longer effective timescales, with the three-factor gate and RMS providing smaller relative gains.

### E.1 SYNAPSE-SPECIFIC VS FACTORIZED ELIGIBILITY TRACES

To quantify the trade-off between synapse-specific $O(N^2)$ traces and neuron-level $O(N)$ factorization, we compared the full Online SNN against a variant in which eligibility traces were stored as per-neuron vectors and weight updates were reconstructed as outer products at the end of a sequence. On 10 Zenodo sessions, both the True Online and Batched Online SNNs with synapse-specific traces significantly outperformed the factorized variant on both X and Y correlations under one-sided Welch t-tests (Table 6), with average gains of roughly 0.06–0.08 in $R$.

Table 6: Comparison between synapse-specific $O(N^2)$ eligibility traces and neuron-level $O(N)$ factorization on Zenodo (10 sessions). Means are shown as mean $\pm$ SEM; $p$-values correspond to one-sided Welch t-tests for $H_1$: $O(N^2)$ performance $> O(N)$ performance.

| Mode | Axis | $O(N^2)$ **mean** | $O(N)$ **mean** | $\Delta R$ | $t$-**statistic** | $p$-**value** |
|------|------|------|------|------|------|------|
| True Online | X | $0.6125 \pm 0.0917$ | $0.5374 \pm 0.0836$ | 0.0751 | 1.91 | 0.036 |
| True Online | Y | $0.7139 \pm 0.0534$ | $0.6571 \pm 0.0496$ | 0.0568 | 2.46 | 0.012 |
| Batched Online | X | $0.6138 \pm 0.0478$ | $0.5374 \pm 0.0836$ | 0.0764 | 2.51 | 0.012 |
| Batched Online | Y | $0.7245 \pm 0.0778$ | $0.6571 \pm 0.0496$ | 0.0674 | 2.31 | 0.018 |

## F OVERVIEW OF SPIKING NEURAL NETWORKS

Spiking Neural Networks (SNNs) represent a class of artificial neural networks that more closely emulate the computational principles of biological nervous systems. Unlike traditional artificial neurons that process continuous-valued activations, SNNs employ discrete, event-driven spikes for information transmission and processing. This bio-mimicry is a primary reason for their selection in BCI research, as it offers a potential avenue for energy-efficient computation, especially when deployed on neuromorphic hardware designed to leverage sparse, spike-based communication (Roy et al., 2019; Furber et al., 2014). The canonical Leaky Integrate-and-Fire (LIF) neuron model, a common choice for SNNs due to its computational simplicity and biological relevance, describes the evolution of a neuron's membrane potential $u(t)$ as:

$$\tau_m \frac{du(t)}{dt} = -\big(u(t) - u_{\text{rest}}\big) + R\,I(t),$$

where $\tau_m$ is the membrane time constant, $u_{\text{rest}}$ the resting potential, $R$ the input resistance, and $I(t)$ the synaptic input current. When $u(t)$ exceeds a threshold $u_{\text{th}}$, the neuron emits a spike, and $u(t)$ is reset to $u_{\text{reset}}$. This event-driven nature can significantly reduce computational load if spikes are sparse.

SNNs are often trained using learning rules inspired by neurobiology, such as Spike-Timing-Dependent Plasticity (STDP). STDP is a form of Hebbian learning where the change in synaptic strength (weight) depends on the precise relative timing of pre-synaptic and post-synaptic spikes (Song et al., 2000). Hebbian plasticity, encapsulated by the maxim "cells that fire together, wire together," is a foundational concept in neuroscience, providing a local, activity-dependent, and thus biologically plausible mechanism for synaptic learning (Hebb, 1949; Markram et al., 1997; Bi & Poo, 1998; Caporale & Dan, 2008). Its locality means that weight updates depend only on information available at the synapse, avoiding the need for global error signals or complex backward computations characteristic of algorithms like backpropagation. This makes Hebbian-like rules attractive for online, adaptive systems where computational resources are limited, and biological realism is desired. While pure STDP can learn temporal correlations, its application to supervised tasks like BCI decoding often requires modulation by reward or error signals, leading to three-factor learning rules, which our work builds upon. The challenge with direct application of traditional spike-based learning rules on conventional hardware can be the substantial computational and memory overhead for tracking precise spike times and managing complex plasticity dynamics, motivating our development of an efficient, online SNN learning algorithm.

## G  MEMORY EFFICIENCY ANALYSIS

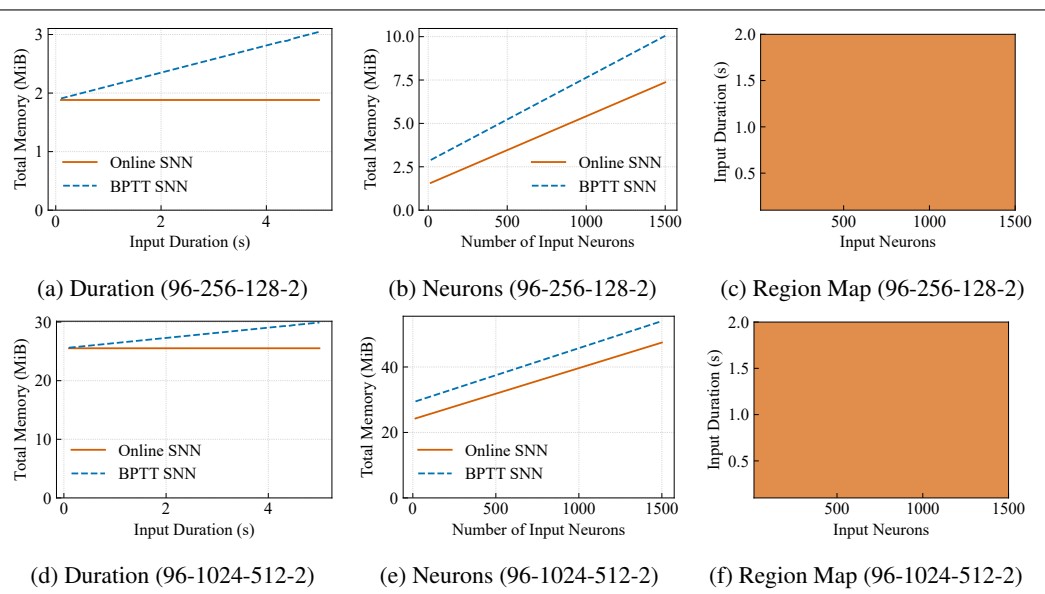

Figure 8: Memory efficiency comparison between Online SNN and BPTT SNN approaches for two representative architectures. Panels (a,d) show memory usage versus input duration, demonstrating the Online SNN's constant-in-$T$ memory requirement versus BPTT's roughly linear scaling. Panels (b,e) show scaling with input dimensionality. Panels (c,f) show operational regions where Online SNN (orange) or BPTT SNN (blue) has lower memory requirements, illustrating that the Online SNN provides increasing absolute savings for longer sequences and wider networks.

Our Online SNN approach has a reduced memory footprint compared to SNNs trained with Backpropagation Through Time (BPTT). This matters for implantable BCIs, which face strict power and memory constraints. The fundamental difference lies in how each approach handles memory allocation during training and inference.

**Memory Calculation Methodology**  We categorize memory usage into static and dynamic components. Static memory includes all data structures that remain constant regardless of input sequence length, while dynamic memory scales with temporal sequence duration.

For the **Online SNN**, memory requirements consist entirely of static components:

- **Network parameters**: All synaptic weights and biases across the three-layer architecture
- **Eligibility traces**: Two sets of trace matrices (fast and slow) that maintain the same dimensions as the corresponding weight matrices
- **Homeostatic variables**: Running statistics for RMS normalization, which scale with network width rather than sequence length

The total parameter count includes: input-to-hidden connections, recurrent hidden-to-hidden connections, hidden-to-hidden connections between layers, and hidden-to-output connections, plus all associated biases. Each parameter requires storage in single-precision floating-point format (4 bytes). In addition to the parameters $(W + b)$, the implementation maintains three weight-sized buffers per trained matrix (fast trace, slow trace, and a momentum/RMS accumulator), so the dominant static memory scales as approximately four times the base weight count, in agreement with the $(W + b) + 3W$ breakdown discussed in Section 2.2. Temporary Hebbian buffers used inside a timestep are reused across timesteps and are not counted as persistent state.

For the **BPTT SNN**, memory requirements include both static and dynamic components. **Static memory**: Network parameters, gradient buffers (same size as parameters), and optimizer state variables (Adam requires two additional buffers per parameter (Kingma & Ba, 2015)). **Dynamic**

**memory**: Computational graph storage for backpropagation, including intermediate activations (hidden states, membrane potentials), spike outputs, and gradient flow information across the temporal sequence used by automatic differentiation.

The static memory for BPTT scales as four times the base parameter count (parameters, gradients, and two Adam optimizer state buffers). The dynamic memory grows linearly with sequence length, as each timestep requires storage of layer activations, membrane states, spike outputs, and intermediate computational results needed for gradient computation through the temporal unfolding.

**Quantitative Analysis**   We provide detailed analysis for two representative architectures: (a) 96-256-128-2 and (b) 96-1024-512-2, where the format represents input-hidden1-hidden2-output neuron counts. For a standard 120-timestep sequence (corresponding to 1.2 seconds of neural data at 100 Hz), the memory breakdown is shown in Table 1. For the 96-256-128-2 architecture, the Online SNN requires 1.88 MiB while the BPTT SNN requires 3.27 MiB at peak usage; for the 96-1024-512-2 architecture, the Online SNN requires 25.53 MiB compared to BPTT's 30.80 MiB. These short-sequence measurements already reflect the removal of BPTT's $O(T)$ activation term. For longer sequences, the extended analysis in Table 7 shows that the Online SNN's peak memory remains constant while the BPTT SNN's memory grows roughly linearly with $T$, leading to substantial absolute and relative savings as sequence length increases. The dynamic memory component for BPTT, which grows roughly linearly with both sequence length and total network width, is responsible for these widening gaps.

**Scaling Properties**   The memory advantage of the Online SNN becomes more pronounced for longer sequences and wider networks, as illustrated in Figure 8 and Tables 7–8. While BPTT memory usage increases linearly with input duration due to the growing computational graph, Online SNN memory remains constant regardless of sequence length. This fixed memory characteristic makes the Online SNN suitable for continuous, long-duration BCI operation where sequences can extend for minutes or hours without interruption. At the same time, varying the hidden size from 32 to 4096 neurons shows that although both methods incur $O(n^2)$ static weight storage, the absolute savings from removing the activation graph grow from a few megabytes to over 170 MB, and relative reductions remain high (roughly 45–97%).

Overall, for typical BCI applications involving continuous neural decoding over extended periods, the Online SNN's fixed-in-$T$ training memory provides advantages over BPTT-based approaches. This memory efficiency, combined with avoiding computational graph storage and gradient accumulation, enables deployment on resource-constrained neuromorphic hardware platforms designed for implantable BCI systems  (Basu et al., 2022; Hasler & Basu, 2025).

Table 7: Peak memory usage (MB) vs. sequence length $T$ for the Online SNN and BPTT-SNN on Zenodo (96-256-128-2) and MC Maze (96-1024-512-2). Ratios and reductions quantify the widening gap as $T$ increases.

| Dataset | Hidden | SeqLen | Online (MB) | BPTT Peak (MB) | Ratio | Reduction (%) | Savings (MB) |
|---|---|---|---|---|---|---|---|
| Zenodo | 256 | 5 | 1.88 | 1.88 | 1.03 | 3.1 | 0.06 |
| MC Maze | 1024 | 5 | 25.53 | 25.76 | 1.01 | 0.9 | 0.23 |
| **Zenodo** | **256** | **50** | **1.88** | **2.46** | **1.31** | **23.7** | **0.58** |
| **MC Maze** | **1024** | **50** | **25.53** | **27.73** | **1.09** | **7.9** | **2.20** |
| **Zenodo** | **256** | **500** | **1.88** | **7.69** | **4.08** | **75.5** | **5.80** |
| **MC Maze** | **1024** | **500** | **25.53** | **47.45** | **1.86** | **46.2** | **21.93** |
| **Zenodo** | **256** | **5000** | **1.88** | **59.89** | **31.82** | **96.9** | **58.01** |
| **MC Maze** | **1024** | **5000** | **25.53** | **244.67** | **9.58** | **89.6** | **219.15** |

# H   BACKGROUND ON LEARNING RULES AND PLASTICITY

## H.1   THREE-FACTOR LEARNING RULES AND ELIGIBILITY TRACES

Modern biologically plausible learning in SNNs builds upon classical Hebbian plasticity but extends it with three-factor learning rules that incorporate an additional modulatory signal  (Frémaux & Gerstner, 2016; Kusmierz et al., 2017; Seung, 2003; Izhikevich, 2007; Florian, 2007; Mozafari et al.,

Table 8: Peak memory usage (MB) vs. hidden-layer width $N$ for a family of 96-$N$-$N/2$-2 SNNs on a fixed sequence length. Online SNN memory grows with $O(N^2)$ static parameters, but BPTT's dynamic activation memory causes total peak usage to rise much faster, leading to larger absolute and relative savings.

| Hidden | Online (MB) | BPTT Peak (MB) | BPTT Dynamic (MB) | Ratio | Reduction (%) | Savings (MB) |
|--------|-------------|----------------|-------------------|-------|---------------|--------------|
| 32 | 0.07 | 2.27 | 2.20 | 31.93 | 96.9 | 2.20 |
| 64 | 0.19 | 3.73 | 3.54 | 19.73 | 94.9 | 3.54 |
| 128 | 0.57 | 6.80 | 6.23 | 12.01 | 91.7 | 6.23 |
| 256 | 1.88 | 13.49 | 11.60 | 7.17 | 86.0 | 11.60 |
| 512 | 6.76 | 29.11 | 22.34 | 4.30 | 76.8 | 22.35 |
| 1024 | 25.53 | 69.37 | 43.83 | 2.72 | 63.2 | 43.84 |
| 2048 | 99.05 | 185.87 | 86.80 | 1.88 | 46.7 | 86.82 |
| 4096 | 390.11 | 562.89 | 172.73 | 1.44 | 30.7 | 172.78 |

2018). Classical spike-timing-dependent plasticity (STDP) represents a two-factor Hebbian rule based on pre- and postsynaptic spike coincidences. However, three-factor learning rules incorporate a third factor—such as a neuromodulator, reward signal, or error feedback—that provides global context to guide synaptic changes. This third factor enables superior credit assignment and adaptation in SNNs, bridging the gap between neuroscience and machine learning perspectives (Mazurek et al., 2025).

A key innovation for biologically plausible credit assignment is the introduction of *eligibility traces*—transient memory variables that accumulate local pre/post activity influence over time (Bellec et al., 2020; Zenke & Ganguli, 2018). These traces decay like synaptic "footprints" of recent activity and are later modulated by a third factor (error or reward) to effect weight changes. This forms the basis of algorithms like e-prop (eligibility propagation) and SuperSpike. Bellec et al. demonstrated that e-prop, which uses ODE-inspired decaying traces, can train recurrent SNNs online with performance close to backpropagation-through-time (Bellec et al., 2020). Similarly, Zenke & Ganguli's SuperSpike derived a voltage-based three-factor rule for multilayer SNNs, showing that purely local spike traces combined with a global error signal can train multilayer networks (Zenke & Ganguli, 2018).

Crucially, these approaches perform online updates: the network does not need to be unfolded in time, avoiding the high memory costs associated with BPTT. The role of eligibility traces is supported by neuroscience—neurons are believed to maintain biochemical traces of pre/post spike pairings that remain dormant until a top-down signal (e.g., dopamine burst) arrives (Frémaux & Gerstner, 2016; Seung, 2003; Izhikevich, 2007). In effect, surrogate gradient algorithms with eligibility traces implement biologically inspired credit assignment: local Hebbian accumulation of potential changes, gated by a global factor when a learning signal occurs.

## H.2 DUAL TIMESCALES

Biological synapses exhibit multiple timescales of plasticity—fast changes that capture recent events and slower processes for long-term stability (Benna & Fusi, 2016; Zenke et al., 2017). Many SNN learning models mirror this by employing dual timescale traces or learning rates. For instance, modern implementations maintain both "fast" and "slow" eligibility traces, reminiscent of the triplet STDP model that used separate decay constants for different spike interactions (Pfister & Gerstner, 2006). Neuromorphic hardware platforms like Intel's Loihi chip explicitly support this architecture, offering multiple filtered spike traces per synapse with short and long time constants, enabling rules that combine precise spike-timing effects with longer-term rate-based effects (Davies et al., 2018).

The interplay of fast and slow components enables continual learning without catastrophic forgetting. Benna and Fusi's synaptic memory model argued that coupling fast and slow weight variables can greatly extend memory lifetime (Benna & Fusi, 2016). Likewise, research has shown that fast updates combined with slower weight stabilization mechanisms improve continual learning (Zenke et al., 2017; Kirkpatrick et al., 2017).

# I   ABLATION STUDY DESIGN

To evaluate each component's contribution within the Online SNN's learning rule, we conduct several ablation studies. These compare the full model to variants where specific components are removed or modified. The experimental setup for these studies is otherwise kept consistent with the main offline evaluation.

This study isolates the contribution of the three-factor learning mechanism by comparing the full Online SNN with a variant where the neuronal sensitivity term is removed. Specifically, we compare the full three-factor rule (Equation 1) against a simplified delta rule that eliminates the surrogate gradient term:

$$\Delta W_{ij}^{\text{delta}}(t) = \mathbf{e}_j(t)\,\text{pre}_i(t) \qquad\qquad \text{(Delta Rule)}$$

This delta rule ablation removes the neuronal sensitivity gating ($d_{\text{LIF}}$) while maintaining the error-driven and activity-dependent components. The update rule in this ablated version effectively reduces to a basic error-driven change, proportional only to the error signal without local neuron sensitivity modulation. This ablation assesses the importance of recurrent connections within the SNN; the full recurrent network is compared to a feedforward-only variant, where the recurrent input to the first hidden layer is removed, and information flows strictly in one direction. Furthermore, we examine the role of different temporal integration windows for the eligibility traces, which accumulate weight updates over time, by comparing the full model's dual-timescale traces (combining fast and slow decay components) against variants using only a fast timescale (shorter memory), only a slow timescale (longer memory), or a single intermediate timescale. Finally, we investigate the effect of normalizing error signals and neuronal activities by their root-mean-square (RMS) values during the update process, comparing the full model with RMS normalization against versions with partial RMS normalization (e.g., only on the output error) and no RMS normalization at all. In all timescale ablations, the base trace constants are fixed at $\tau_{\text{fast}} = 120$ ms and $\tau_{\text{slow}} = 700$ ms, and the medium condition uses a single intermediate constant of 400 ms. Fast-only, slow-only, and medium-only configurations disable the other trace and set the mixing coefficient so that the active trace carries the entire eligibility history, without changing any other part of the update rule. For true online runs we use slightly shorter effective trace constants (scaling $\tau_{\text{fast}}$ and $\tau_{\text{slow}}$ as described in Appendix D) and in the dual condition we retain both traces but adjust the fixed mixture weight to put more emphasis on the fast trace in the online setting and an equal mixture in the batched setting.

For ablation analysis and statistical testing, each configuration produces per-session Pearson correlations for the X and Y velocity components. For plotting, the aggregate statistic shown for each configuration and axis is the across-session mean, with error bars reflecting a normal-approximation 95% confidence interval computed as $\pm 1.96$ times the standard error of the mean, where the standard deviation uses the unbiased estimator with degrees of freedom equal to one less than the number of sessions. Significance annotations are derived from paired, two-sided t-tests across sessions that compare a designated baseline configuration to each alternative within the same ablation family, performed separately for X and for Y. Pairing is enforced by aligning on the intersection of sessions that contain measurements for both configurations; comparisons with fewer than two paired observations are omitted. Significance thresholds use the following convention: p < 0.001 is annotated with three stars, p < 0.01 with two stars, p < 0.05 with one star, and p $\geq$ 0.05 is marked as not significant. No multiple-comparisons correction is applied, as tests are presented as descriptive indicators alongside effect direction inferred from the plotted means. Baselines are chosen per family: for trace timescale comparisons, a canonical ordering defines which configuration is treated as the baseline; for update-mechanism comparisons, the dual-update configuration serves as the baseline. When update-mechanism analyses require a dual baseline that is not present in the current results table, the dual baseline is drawn from the corresponding trace-timescale results and paired only on sessions common to both sources, preserving the within-session design. Configuration labels may be harmonized to canonical names before analysis to ensure consistent grouping across files. The visualizations present grouped bars for X and Y per configuration on a common vertical scale from zero to slightly above one to include the confidence intervals.

## I.1   HYPERPARAMETER SENSITIVITY

We also performed a sensitivity study on five randomly selected Zenodo sessions to assess robustness to temporal and optimization hyperparameters. Table 9 summarizes the ranges explored, baseline

settings, and relative performance spread; no configuration differed significantly from baseline under Welch's t-tests, and the biologically motivated temporal parameters showed the narrowest spread despite wide variation ranges.

Table 9: Hyperparameter sensitivity on Zenodo (five sessions). Spread is the percentage range of mean correlation across tested values. All configurations showed no statistically significant difference from baseline (Welch's t-tests, all $p > 0.49$).

| Hyperparameter | Values tested | Baseline | Spread (%) | Best config | $p$-value | Significant? |
|---|---|---|---|---|---|---|
| $\tau_{\text{fast}}$ (ms) | 60, 120, 180 | 120 | 1.4 | 180/700 | 0.88 | No |
| $\tau_{\text{slow}}$ (ms) | 400, 700, 1000 | 700 | 1.4 | 180/700 | 0.88 | No |
| $\alpha_{\text{mix}}$ | 0.1, 0.3, 0.5, 0.7, 0.9 | 0.5 | 1.8 | 0.5 | baseline | No |
| fast_lr | $1\times10^{-4}$ to $5\times10^{-3}$ (5 values) | $2\times10^{-3}$ | 10.6 | $2\times10^{-3}/5\times10^{-5}$ | 0.93 | No |
| slow_lr | $1\times10^{-5}$ to $5\times10^{-4}$ (5 values) | $2\times10^{-4}$ | 10.6 | $2\times10^{-3}/5\times10^{-5}$ | 0.93 | No |
| window_size | 10, 25, 50, 100, 200 | 50 | 1.2 | 100 | 0.95 | No |

## J   CLOSED-LOOP SIMULATION METHODS

To further evaluate the adaptive capabilities of the Online SNN, particularly in dynamic environments, we conducted a series of online closed-loop BCI simulations using synthetically generated neural data. Closed-loop simulation is an important validation approach for BCI systems, as demonstrated by Cunningham et al. (Cunningham et al., 2011), who showed that offline analyses can be misleading and that closed-loop testing reveals fundamentally different optimal parameters than offline evaluation. Their work demonstrated that while offline analyses suggested optimal Kalman filter bin widths of 100-300ms, closed-loop testing with human subjects revealed that much shorter bin widths (25-50ms) yielded superior decode performance when feedback control was incorporated. This finding was subsequently confirmed in closed-loop rhesus monkey experiments, establishing that closed-loop simulation is necessary for understanding the role of feedback control and co-adaptation between the user and decoder (Cunningham et al., 2011). These simulations model how decoder outputs continuously influence the system state and subsequent neural inputs. This reflects the sequential dependencies and error characteristics in real BCI operation that cannot be captured through static offline analysis. This section details the methodology for these simulations. We investigate two distinct closed-loop simulation protocols: one for assessing adaptation to neural disruptions and another for evaluating online learning from scratch.

### J.0.1   SYNTHETIC NEURAL DATA GENERATION

We simulate a 2D cursor control task using a synthetic neural population comprising 96 neurons with cosine-tuned directional preferences. Each neuron $i$ has a preferred direction $\mathbf{d}_i \in \mathbb{R}^2$ with unit norm, distributed uniformly across four quadrants to ensure balanced directional representation.

Given a target velocity $\mathbf{v}(t) \in \mathbb{R}^2$ at timestep $t$, the firing rate for neuron $i$ is computed as:

$$r_i(t) = r_{\min} + (r_{\max} - r_{\min}) \cdot \max\left(0, \frac{\mathbf{d}_i \cdot \hat{\mathbf{v}}(t) + 0.5}{1.5}\right) \tag{7}$$

where $\hat{\mathbf{v}}(t) = \mathbf{v}(t)/(2.0 \cdot \|\mathbf{v}(t)\|)$ is the scaled velocity, $r_{\min} = 5$ Hz and $r_{\max} = 100$ Hz are the minimum and maximum firing rates, respectively. When $\|\mathbf{v}(t)\| = 0$, we set $\hat{\mathbf{v}}(t) = \mathbf{0}$ to avoid division by zero.

The spike probability for neuron $i$ is then:

$$p_i(t) = \text{clamp}(r_i(t) \cdot \Delta t + \mathcal{N}(0, \sigma^2), 0, 1) \tag{8}$$

where $\Delta t = 0.01$ s is the timestep duration, $\sigma = 0.02$ is the noise standard deviation, and $\mathcal{N}(0, \sigma^2)$ represents Gaussian noise. Finally, binary spikes are generated as $s_i(t) \sim \text{Bernoulli}(p_i(t))$.

### J.0.2   TASK STRUCTURE FOR CLOSED-LOOP SIMULATION

The experimental paradigm involves a sequence of 2D center-out reach tasks on a simulated screen of dimensions 800x600 units. At each trial, a target is positioned at coordinates $(x_{\text{target}}, y_{\text{target}})$ with

distance constraint $||\mathbf{p}_{\text{target}} - \mathbf{p}_{\text{center}}|| > 3R$ from the center position $\mathbf{p}_{\text{center}} = [400, 300]^T$, where $R = 50$ units is the target radius.

The closed-loop control dynamics operate as follows. At each timestep $t$, the desired velocity is computed using a proportional controller:

$$\mathbf{v}_{\text{desired}}(t) = \frac{\mathbf{p}_{\text{target}} - \mathbf{p}_{\text{cursor}}(t)}{||\mathbf{p}_{\text{target}} - \mathbf{p}_{\text{cursor}}(t)||} \cdot \min\left(1.0, \frac{||\mathbf{p}_{\text{target}} - \mathbf{p}_{\text{cursor}}(t)||}{200}\right) \qquad (9)$$

where $\mathbf{p}_{\text{cursor}}(t)$ is the current cursor position. This desired velocity drives the synthetic neural population to generate spikes $\mathbf{s}(t)$ according to Equations (7-8).

The decoder processes these spikes to produce an estimated velocity $\hat{\mathbf{v}}_{\text{decoder}}(t)$, which updates the cursor position:

$$\mathbf{p}_{\text{cursor}}(t+1) = \text{clamp}\left(\mathbf{p}_{\text{cursor}}(t) + \alpha \cdot \hat{\mathbf{v}}_{\text{decoder}}(t), [0, 800] \times [0, 600]\right) \qquad (10)$$

where $\alpha = 5$ is the movement scaling factor. Success occurs when $||\mathbf{p}_{\text{target}} - \mathbf{p}_{\text{cursor}}(t)|| < R$ within 300 timesteps (3 seconds).

### J.0.3 EXPERIMENTAL PROTOCOLS AND PHASES

**Online Adaptation and Disruption Ablations**   Each experimental run in this protocol is structured into two phases to assess performance before and after a controlled neural perturbation. Decoders are evaluated in matched conditions but not with identical spike sequences at the same timestamps.

**Phase 1: Initial Learning (Pre-Disruption):** All decoders (KF, LSTM, BPTT SNN) first undergo an initial calibration phase using a dataset of 10,000 synthetically generated spike–velocity steps. The KF parameters are estimated via maximum likelihood, while LSTM and BPTT SNN undergo offline training.

This dataset was generated using a closed-loop methodology to better reflect interactive BCI dynamics. An open-loop alternative (spikes from idealized trajectories) was considered but does not capture how decoder outputs influence future inputs and errors.

To enhance ecological validity, the 10,000-step dataset for calibrating the Kalman Filter, LSTM, and BPTT SNN was generated in closed loop. An untrained LSTM drove the cursor; at each timestep, the ideal velocity to target was recorded as the label, spikes were generated from that ideal velocity, and the untrained LSTM's output advanced the cursor. This yielded data collected under decoder-driven trajectories with realistic sequential dependencies. The KF parameters were estimated from this dataset, while LSTM and BPTT SNN were trained in a supervised manner before evaluation.

In this protocol, the Online SNN was pre-trained through 100 closed-loop reach attempts to establish an initial functional state. Following this training, all decoders were evaluated over 150 successful reach trials using the unperturbed spike generator with baseline noise and tuning.

**Phase 2: Adaptation to Disruption:** Immediately following Phase 1, a specific disruption is introduced to the spike generator's properties. Three types of disruptions are investigated in separate experiments:

**Remapping:** For intensity $\lambda \in [0, 1]$, a fraction $\min(0.95, 1.9\lambda)$ of neurons have their preferred directions reassigned. For selected neuron $i$, new preferred directions are computed as:

$$\mathbf{d}_i^{\text{new}} = [\cos(\theta_i^{\text{new}}), \sin(\theta_i^{\text{new}})]^T \qquad (11)$$

where $\theta_i^{\text{new}} = \text{Uniform}(0, 2\pi) + \text{Bernoulli}(0.5) \cdot \pi$ to create dramatic directional shifts.

**Drift:** Firing rate parameters are modified as:

$$r_{\max}^{\text{new}} = r_{\max}^{\text{orig}} \cdot (1.0 - 1.6\lambda) \qquad (12)$$

$$r_{\min}^{\text{new}} = r_{\min}^{\text{orig}} \cdot (1.0 + 6\lambda) \qquad (13)$$

Additionally, noise variance increases with a capped schedule: $\sigma^{\text{new}} = \min(0.4, \sigma^{\text{orig}} \cdot (1 + 4\lambda))$.

**Dropout:** A fraction $\min(0.9, 1.8\lambda)$ of neurons are silenced by applying a binary mask $\mathbf{m} \in \{0, 1\}^{96}$ to the spike output: $\mathbf{s}_{\text{masked}}(t) = \mathbf{s}(t) \odot \mathbf{m}$.

After disruption onset, all decoders were evaluated on an additional 50 reach trials. During this phase, only the Online SNN continued to adapt its weights online; the KF, LSTM, and BPTT SNN operated with fixed weights from the initial offline training.

**No-Pretrain / Online-Only Calibration Comparison** To investigate decoder capability to learn from scratch and compare online versus offline calibration, a separate experimental protocol is employed. In **Phase 1: Online Data Collection & Online SNN Adaptation**, all decoders (KF, LSTM, BPTT SNN, Online SNN) are initialized with random weights (or default parameters for KF). Each decoder then controls the cursor for 30 reach attempts. During these trials, the Online SNN actively adapts its weights based on ongoing performance. The other decoders (KF, LSTM, BPTT SNN) operate with fixed random weights, solely for the purpose of collecting a diverse set of spike-velocity data under their respective (initially poor) control policies. Data from each decoder's interaction (spike trains and ideal velocities) are stored separately. Following Phase 1, in **Phase 2: Post-Calibration Evaluation**, the offline-trainable decoders (KF, LSTM, BPTT SNN) are trained (calibrated) using only the data they respectively collected during their 30 interaction trials in Phase 1. The Online SNN carries forward the weights learned during its Phase 1 online adaptation. Subsequently, all four decoders are evaluated on a new set of 70 reach trials.

### J.0.4 OFFLINE TRAINING HYPERPARAMETERS

### J.0.5 EVALUATION METRICS FOR CLOSED-LOOP SIMULATIONS

Decoder performance is primarily quantified by **Time to Reach Target (seconds)**- For each successful reach, the number of 10ms timesteps taken to acquire the target is recorded. This is converted to seconds for reporting. Timeouts (reaching the 3-second limit) are assigned the maximum duration.

The **Mean and Standard Error of the Mean (SEM) of Reach Times** are computed across 10 independent experimental runs, each initiated with a different random seed, to account for stochasticity in target placement, spike generation, and model initializations. Learning curves plot the smoothed (rolling mean, window N=15) time to reach target across consecutive trials.

### J.0.6 IMPLEMENTATION DETAILS FOR CLOSED-LOOP SIMULATIONS

All experiments are conducted using custom simulation scripts leveraging PyTorch, `snnTorch`, NumPy, and Matplotlib. The simulation environment is structured to ensure reproducibility, with controlled random seeds for all stochastic processes. The synthetic environment and decoder evaluations are performed within a consistent simulation loop ensuring fair comparisons. Decoders are evaluated separately in individual reach attempts, not concurrently with identical spike data.

## K  ADDITIONAL BASELINE COMPARISONS

Table 10: Complete hyperparameters for offline SNN training on MC Maze and Zenodo datasets.

| Parameter | MCMaze Batched Online SNN | MCMaze True Online SNN | Zenodo Batched Online SNN | Zenodo True Online SNN |
|---|---|---|---|---|
| *Architecture* | | | | |
| Input size | 182 | 182 | 96 | 96 |
| Hidden layer 1 | 1024 | 1024 | 256 | 256 |
| Hidden layer 2 | 512 | 512 | 128 | 128 |
| Output size | 2 | 2 | 2 | 2 |
| Recurrent connections | Yes (layer 1) | Yes (layer 1) | Yes (layer 1) | Yes (layer 1) |
| *Data Processing* | | | | |
| Bin width | 100ms | 100ms | 50ms | 50ms |
| Stride | 10ms | 10ms | 50ms | 50ms |
| Kinematic lag | 80ms | 80ms | 0ms | 0ms |
| Spike processing | Count | Count | Count | Count |
| Normalization | None | None | None | None |
| *Neuron Parameters* | | | | |
| Surrogate gradient | Fast sigmoid | Fast sigmoid | Fast sigmoid | Fast sigmoid |
| LIF $\beta$ (layers 1,2) | 0.7, 0.7 | 0.7, 0.7 | 0.7, 0.7 | 0.7, 0.7 |
| LIF $\beta$ (output) | 0.5 | 0.5 | 0.5 | 0.5 |
| LIF threshold | 1.0 | 1.0 | 1.0 | 1.0 |
| Reset mechanism | None (output) | None (output) | None (output) | None (output) |
| *Learning Parameters* | | | | |
| Fast learning rate | $1 \times 10^{-5}$ | $1 \times 10^{-4}$ | $3 \times 10^{-3}$ | $2 \times 10^{-3}$ |
| Slow learning rate | $1 \times 10^{-6}$ | $1 \times 10^{-5}$ | $1 \times 10^{-3}$ | $2 \times 10^{-4}$ |
| $\tau_{\text{fast}}$ | 120ms | 120ms | 120ms | 120ms |
| $\tau_{\text{slow}}$ | 700ms | 700ms | 700ms | 700ms |
| Trace mixing $\alpha_{\text{mix}}$ | 0.5 | 0.8 | 0.5 | 0.8 |
| Window size $K$ | 50 | 200 | 50 | 50 |
| Momentum $\mu$ | 0.9 | 0.9 | 0.9 | 0.9 |
| Weight decay | $1 \times 10^{-5}$ | $1 \times 10^{-5}$ | $1 \times 10^{-5}$ | $1 \times 10^{-5}$ |
| Weight cap $c_\ell$ | 6.0 | 6.0 | 6.0 | 6.0 |
| *Training* | | | | |
| Epochs (disregarding early stopping) | 20 | 3 (due to high data volume for single timestep) | 20 | 20 (low data volume for single timestep) |
| Batch size | 32 | 1 | 32 | 1 |
| Sequence length | 10 | 1 | 10 | 1 |
| Patience | 10 | 8 | 10 | 20 |
| Eval frequency | each epoch | 1000 steps | each epoch | 500 steps |
| State reset | trial bound. | trial bound. | N/A | 200 steps |

Table 11: Hyperparameters for baseline decoders (LSTM and Kalman Filter) on MC Maze and Zenodo datasets.

| Parameter | MC Maze LSTM | Zenodo LSTM | MC Maze KF | Zenodo KF |
|---|---|---|---|---|
| *Architecture* | | | | |
| Input size | 182 | 96 | 182 | 96 |
| Hidden units | 256 | 256 | N/A | N/A |
| Layers | 2 | 2 | N/A | N/A |
| Output size | 2 | 2 | 2 | 2 |
| Dropout | 0.2 | 0.2 | N/A | N/A |
| *Data Processing* | | | | |
| Spike processing | Rate (Hz) | Rate (Hz) | Rate (Hz) | Rate (Hz) |
| Normalization | Z-score | Z-score | Z-score | Z-score |
| Sequence length | 10 (offline), 5 (online) | 10 (offline), 5 (online) | N/A | N/A |
| *State Space (KF only)* | | | | |
| State dimensions | N/A | N/A | 2 (velocity) | 2 (velocity) |
| Observation dim. | N/A | N/A | 182 | 96 |
| Process noise Q | N/A | N/A | 0.1 | 0.1 |
| Observation noise R | N/A | N/A | 1.0 | 1.0 |
| Initial covariance P | N/A | N/A | 100.0 | 100.0 |
| *Training (Offline)* | | | | |
| Epochs (disregarding early stopping) | 50 | 50 | N/A (analytical) | N/A (analytical) |
| Batch size | 64 | 64 | N/A | N/A |
| Learning rate | $1 \times 10^{-3}$ | $1 \times 10^{-3}$ | N/A | N/A |
| Optimizer | Adam | Adam | N/A | N/A |
| Weight decay | $1 \times 10^{-5}$ | $1 \times 10^{-5}$ | N/A | N/A |
| Patience | 10 | 10 | N/A | N/A |

Table 12: Hyperparameters for Zenodo ANN baselines (MLP and GRU) used in the additional baseline comparison.

| Parameter | Zenodo MLP | Zenodo GRU |
|---|---|---|
| *Architecture* | | |
| Input size | 96 | 96 |
| Hidden units | 256 | 256 |
| Layers | 2 (fully connected) | 2 (GRU layers) |
| Output size | 2 | 2 |
| Dropout | 0.0 | 0.0 |
| *Data Processing* | | |
| Spike processing | Rate (Hz) | Rate (Hz) |
| Normalization | Z-score | Z-score |
| Sequence length | Full session | Full session |
| *Training* | | |
| Epochs | 50 | 50 |
| Batching | Single-sequence | Single-sequence |
| Learning rate | $1 \times 10^{-3}$ | $5 \times 10^{-4}$ |
| Optimizer | Adam | Adam |
| Weight decay | $1 \times 10^{-5}$ | $1 \times 10^{-5}$ |

Table 13: Complete hyperparameters for closed-loop experiments across all protocols and decoders.

| Parameter | Adaptation Protocol | No-Pretrain Protocol | BCI Simulation Default Parameters |
|---|---|---|---|
| *Neural Population & Environment* | | | |
| Neural population | 96 neurons | 96 neurons | 96 neurons |
| Spike generator noise | 0.02 | 0.02 | 0.02 |
| Screen dimensions | 800×600 units | 800×600 units | 800×600 units |
| Target radius | 50 units | 50 units | 50 units |
| Center position | [400, 300] | [400, 300] | [400, 300] |
| Movement scale | 5 units/step/vel | 5 units/step/vel | 5 units/step/vel |
| Max reach duration | 3.0 seconds | 3.0 seconds | 10.0 seconds |
| Disruption intensities | Remap/Drift/Dropout (0.9) | N/A | Remap/Drift/Dropout (0.5) |
| *Task Structure* | | | |
| Offline pretraining steps | 10,000 | N/A | 10,000 |
| Online SNN pretrain reaches | 100 | N/A | 400 |
| Initial learning reaches | 150 | N/A | 400 |
| Adaptation reaches | 50 | N/A | 120 |
| Online data collection | N/A | 30 reaches | N/A |
| Post-calibration eval | N/A | 70 reaches | N/A |
| *SNN Architecture (All Variants)* | | | |
| Input size | 96 | 96 | 96 |
| Architecture | 96-256-128-2 | 96-256-128-2 | 96-512-256-2 |
| Hidden LIF $\beta$ | 0.7, 0.7 | 0.7, 0.7 | 0.7, 0.7 |
| Output LIF $\beta$ | 0.5 | 0.5 | 0.5 |
| LIF threshold | 1.0 | 1.0 | 1.0 |
| Output reset | None | None | None |
| Surrogate gradient | Fast sigmoid | Fast sigmoid | Fast sigmoid |
| *LSTM Architecture* | | | |
| Input size | 96 | 96 | 96 |
| Architecture | 96-256-256-2 | 96-256-256-2 | 96-256-256-2 |
| Layers | 2 | 2 | 2 |
| Dropout | 0.2 | 0.2 | 0.2 |
| *Kalman Filter* | | | |
| State dimensions | 2 (velocity) | 2 (velocity) | 4 (pos+vel) |
| Observation dim | 96 | 96 | 96 |
| Process noise Q | 0.1 | 0.1 | 0.1 |
| Observation noise R | 1.0 | 1.0 | 1.0 |
| Initial covariance P | 100.0 | 100.0 | 100.0 |
| *Online SNN Learning* | | | |
| Fast learning rate | $1 \times 10^{-4}$ | $1 \times 10^{-4}$ | $1 \times 10^{-4}$ |
| Slow learning rate | $1 \times 10^{-3}$ | $1 \times 10^{-3}$ | $1 \times 10^{-3}$ |
| Window size K | 10 | 10 | 10 |
| $\tau_{\text{fast}}$ | 120ms | 120ms | 120ms |
| $\tau_{\text{slow}}$ | 700ms | 700ms | 700ms |
| Trace mixing $\alpha_{\text{mix}}$ | 0.5 | 0.8 | 0.5 |
| Momentum $\mu$ | 0.9 | 0.9 | 0.9 |
| Weight cap $c_\ell$ | 6.0 | 6.0 | 6.0 |
| Weight decay | $1 \times 10^{-5}$ | $1 \times 10^{-5}$ | $1 \times 10^{-5}$ |
| *Offline Training (LSTM & BPTT-SNN)* | | | |
| Epochs | 30 | 30 | 30 |
| Batch size | 64 | 64 | 64 |
| Sequence length | 50 | 5 | 50 |
| Learning rate | $1 \times 10^{-3}$ | $1 \times 10^{-3}$ | $1 \times 10^{-3}$ |
| Optimizer | Adam | Adam | Adam |
| Weight decay | $1 \times 10^{-5}$ | $1 \times 10^{-5}$ | $1 \times 10^{-5}$ |
| Patience | 10 | 5 | 10 |

Table 14: Zenodo baseline comparison (ten sessions). Average correlation, online $T{=}1$ capability, optimizer requirement, and hardware considerations for each method, highlighting an integer-friendly, neuromorphic-compatible but as yet unmeasured hardware profile for the proposed SNN.

| Method | Avg corr | Online ($T{=}1$) | Optimizer | Hardware |
|---|---|---|---|---|
| Kalman | $0.44 \pm 0.06$ | Yes | None (closed-form) | CPU-friendly |
| MLP | $0.39 \pm 0.15$ | No | Adam | CPU/GPU (no temporal state) |
| **SNN (ours)** | **$0.66 \pm 0.05$** | **Yes** | **None** | **Integer-friendly, neuromorphic-compatible (hardware measurements future work)** |
| GRU | $0.76 \pm 0.05$ | No | Adam + BPTT | GPU/batched |
| LSTM | $0.79 \pm 0.04$ | No | Adam + BPTT | GPU/batched |

Table 15: Comparison to recent SNN intracortical decoder works. Our method emphasizes dual-timescale Hebbian accumulation, integer-friendly operations aimed at neuromorphic compatibility, and explicit non-stationarity evaluation on intracortical datasets, while leaving direct hardware measurements to future work.

| Method | Online $T{=}1$ | Dual timescale | Integer-friendly | Performance | Dataset |
|---|---|---|---|---|---|
| Taeckens & Shah (2023) (Taeckens & Shah, 2024) | Yes | No | No | $\rho{\approx}0.70$ | Primate premotor (regression) |
| MotorSRNN (2023) (Liu et al., 2023) | No | No | No | classification acc. | Monkey M1 (classification) |
| LSS-CA-SNN (2024) (Fu et al., 2024) | No | No | No | $R^2{=}0.50\text{--}0.67$ | Rhesus (trajectory regression) |
| Adaptive Pruning (2025) (Rivelli et al., 2024) | No | No | Partial | $R^2{=}80\text{--}98\%$ | NeuroBench NHP |
| **Ours** | **Yes** | **Yes** | **Yes** | **$\rho{=}0.67\text{--}0.82$** | Zenodo, MC Maze (regression) |

