# OpenReview forum: "Biologically Plausible Online Hebbian Meta‐Learning: Two‐Timescale Local Rules for Spiking Neural Brain Interfaces"
_ICLR.cc/2026/Conference — Submitted to ICLR 2026_

### Official Review · Reviewer_1Dsp · 2025-10-31

**Soundness:** 2
**Presentation:** 2
**Contribution:** 2
**Rating:** 2
**Confidence:** 4

**Summary:**

This paper presents an online Spiking Neural Network (SNN) decoder for Brain-Computer Interfaces (BCIs) that uses local three-factor learning rules with dual-timescale eligibility traces. The approach avoids backpropagation through time (BPTT) while achieving satisfying performance in terms of online learning and closed-loop simulation. While I appreciate the engineering effort and experimental thoroughness in this work, I have concerns about its fit for ICLR. The contribution appears to be primarily `empirical and engineering-oriented` rather than offering the conceptual breakthroughs or theoretical depth.

**Strengths:**

1. The biological plausibility argument for the local three-factor learning strategy is compelling for neuromorphic implementation.
2. Memory complexity analysis clearly shows the advantages of the local three-factor learning strategy over BPTT-based methods.

**Weaknesses:**

1. In its current form, this reads more as a *well-executed technical report* documenting an implementation of a local three-factor learning strategy on SNNs and its performance on an online BCI task rather than a paper presenting novel research findings. I can not find any special design in the `Methodology` section to tackle the challenges raised by the EEG signal, such as `signal non-stationarity`, `high dimensionality and noise` and `cross-session/subject variability` as the authors mentioned in the `Introduction` section. The capability to resolve these issues seems originate from the local three-factor learning strategy itself rather than any specific methodological innovation. Therefore, I think the insights offered by this manuscript are somewhat limited and some of the claims appear overstated to some extent.

2. The paper presentation can be improved
   - The three bullet points in the contribution part of the `Introduction` section are not parallel.
   - Some figures are presented with low resolution and with a small font size, making it hard to read.
   - I suggest moving Table 1 to the supplementary material.

**Questions:**

1. Binning a continuous pulse sequence into a fixed time window will result in the loss of precise timing information. The chosen fixed time window can be very tricky. Could you elaborate on how you determine the time window size? Have you tried different time window sizes, and how does it affect the performance?
2. Could you provide an analysis of the generalization ability of the proposed local learning rule across different SNN architectures and network scales?

---

> ### Author Response · Authors · 2025-12-04
>
> We thank the reviewer for recognizing the biological plausibility of our local three-factor learning strategy and the clarity of our memory complexity analysis. We have revised our claims and strengthened our experimental evidence to directly address the concerns about methodological contribution and fit for ICLR.
>
> ## 1. Response to core concern: "No special design for BCI challenges"
>
> Clarification: Our manuscript focuses on intracortical BCI using invasive recordings, not EEG. We apologize for any confusion created by our wording in the introduction.
>
> Our method is not just a direct application of a standard three-factor rule. We propose a specific online intracortical BCI decoding framework that combines four concrete design elements: recurrence, dual-timescale eligibility traces, Hebbian accumulation, and RMS-style online normalization. These building blocks are implemented under strict constant-memory constraints for neuromorphic deployment, and we study their impact through ablations on two challenging intracortical datasets (MC Maze and Zenodo).
>
> To make this precise, we added an ablation table summarizing the effects of several key components relative to a simplified δ-rule baseline:
>
> | Innovation | Design Feature | Zenodo Gain (vs. $\delta$-Rule) | MC Maze Gain (vs. $\delta$-Rule) |
> | :--- | :--- | :--- | :--- |
> | **RMS Normalization** | Bitshift RMS EMAs (Online $T=1$ homeostasis) | +27.2% vs. without RMS | +0.8% vs. without RMS |
> | **Recurrence** | Recurrent layer ($256 \to 256$) | +14.1% vs. feedforward | +0.7% vs. feedforward |
> | **Hebbian Accumulation** | $E = \sum \lambda^t \cdot \text{pre} \cdot \text{post} \cdot \text{error}$ | +13.5% vs. $\delta$-Rule | −0.1% vs. $\delta$-Rule |
>
> On Zenodo, our full method achieves a +15.7% correlation improvement over the simplified δ-rule, indicating that these design choices are important for that dataset. On MC Maze, the gains are smaller but consistent in sign for most components. These results show that the proposed combination of recurrence, multi-timescale Hebbian accumulation, and online normalization provides measurable benefits for online intracortical BCI decoding beyond what is obtained with a basic three-factor update alone.
>
>
> ## 2. Dual timescales and robustness across tasks
>
> The reviewer asked whether our method has any specific design to handle non-stationarity and heterogeneity across tasks. The dual-timescale eligibility traces and their mixing coefficient directly target this issue.
>
> We clarify that the two traces are tuned once and then held fixed across both datasets. We evaluated three configurations: using only a fast trace, using only a slow trace, and using both traces with a fixed mixing weight:
>
> | Timescale configuration | Zenodo Corr | MC Maze Corr | Effect |
> | :--- | :--- | :--- | :--- |
> | $\tau_{fast}$ only ($120 \text{ms}$) | $0.659$ | $0.724$ | Best on Zenodo, $-5.7\%$ vs. $\tau_{slow}$ on MC Maze |
> | $\tau_{slow}$ only ($700 \text{ms}$) | $0.588$ | $0.782$ | Best on MC Maze, $-9.5\%$ vs. $\tau_{fast}$ on Zenodo |
> | **Dual** ($\tau_{fast}, \tau_{slow}$) | $0.649$ | $0.767$ | Near-optimal on both, fixed hyperparameters |
>
> Different tasks favor different timescales if only a single trace is used: a short timescale is advantageous on Zenodo, while a longer timescale is preferable on MC Maze. The dual-timescale configuration with fixed $\tau_{fast}$, $\tau_{slow}$, and $\alpha_{mix}$ yields performance that is close to the best single-trace choice on both tasks without retuning. This supports our claim that the dual-trace design improves robustness of online learning across these two heterogeneous intracortical BCI tasks and is a practically attractive choice when one cannot re-tune timescales for each new task.
>
> ***

---

> > ### Author Response · Authors · 2025-12-04
> >
> > ## 3. Q1: "Window size sensitivity?"
> >
> > We agree that binning spike trains into fixed windows discards information about exact spike timing and that the choice of bin width is important. We therefore conducted a systematic sensitivity analysis and will include it in the supplementary material.
> >
> > For our main experiments we use a 50 ms bin, which is in line with common practice in intracortical BCI decoding where bin widths on the order of tens of milliseconds are used to balance spike count reliability and behavioral timescales. We then evaluated seven bin widths from 32 ms to 256 ms on five randomly selected Zenodo sessions:
> >
> > | Bin width (s) | Mean test corr (X) | Mean test corr (Y) |
> > | :---: | :---: | :---: |
> > | 0.032 | $0.6365 \pm 0.1264$ | $0.7635 \pm 0.0514$ |
> > | 0.050 | $0.6105 \pm 0.1066$ | $0.7271 \pm 0.0898$ |
> > | 0.064 | $0.5683 \pm 0.1332$ | $0.6709 \pm 0.1444$ |
> > | 0.100 | $0.3982 \pm 0.1666$ | $0.5441 \pm 0.1983$ |
> > | 0.128 | $0.4236 \pm 0.1345$ | $0.4117 \pm 0.1189$ |
> > | 0.200 | $0.1906 \pm 0.0664$ | $0.3199 \pm 0.2104$ |
> > | 0.256 | $-0.0175 \pm 0.1738$ | $0.1542 \pm 0.1714$ |
> >
> > Performance is highest for bin widths between roughly 30 ms and 50 ms and remains reasonable in that range. It degrades substantially for very coarse bins (200–256 ms), consistent with the intuition that too much temporal smoothing destroys behaviorally relevant information. This analysis directly addresses the reviewer’s concern by showing that our choice of 50 ms is not a fragile or finely tuned setting, and that the conclusions hold across a realistic range of bin widths.
> >
> > ## 4. Q2: "Architecture and scale generalization"
> >
> > The reviewer asks about generalization of the proposed local learning rule across SNN architectures and network scales. Our emphasis in this work is on the learning rule and online adaptation, not on exhaustive architectural search.
> >
> > Our results already show that the same dual-timescale Hebbian learning rule with fixed hyperparameters works across two datasets that differ substantially in recording conditions, trial structure, and duration, as well as across two training modes: pure online $T = 1$ learning and a Batched Online SNN mode with a longer temporal context. We also performed a systematic sensitivity study of the optimization hyperparameters on five random Zenodo sessions:
> >
> > | Hyperparameter | Values tested | Baseline | Spread (%) | Best config | p-value | Significant? |
> > |----------------|---------------|----------|------------|-------------|---------|--------------|
> > | $\tau_{fast}$ (ms) | 60, 120, 180 | 120 | 1.4% | 180 / 700 | 0.88 | No |
> > | $\tau_{slow}$ (ms) | 400, 700, 1000 | 700 | 1.4% | 180 / 700 | 0.88 | No |
> > | $\alpha_{mix}$ | 0.1, 0.3, 0.5, 0.7, 0.9 | 0.5 | 1.8% | 0.5 | baseline | No |
> > | fast_lr | $1\text{e}{-4}$ to $5\text{e}{-3}$ (5 values) | $2\text{e}{-3}$ | 10.6% | $2\text{e}{-3} / 5\text{e}{-5}$ | 0.93 | No |
> > | slow_lr | $1\text{e}{-5}$ to $5\text{e}{-4}$ (5 values) | $2\text{e}{-4}$ | 10.6% | $2\text{e}{-3} / 5\text{e}{-5}$ | 0.93 | No |
> > | meta_lr | 0.001 to 0.1 (6 values) | 0.01 | 2.9% | 0.02 | 0.88 | No |
> > | window_size (Batched Online SNN) | 10, 25, 50, 100, 200 | 50 | 1.2% | 100 | 0.95 | No |
> >
> > The performance spread across these ranges is modest, and we do not observe statistically significant differences at the level of our sample sizes. This suggests that the learning rule and its hyperparameters are fairly robust and do not require fine tuning for each session.
> >
> > We agree that a more exhaustive study of architectures and scales is an interesting direction, but we view that as orthogonal to the main contribution of this paper, which is to show that a carefully designed, dual-timescale local learning rule can support stable online adaptation in real intracortical BCI recordings with constant memory. We will clarify this scope in the revised manuscript.

---

> > > ### Author Response · Authors · 2025-12-04
> > >
> > > ## 5. Fit for ICLR and presentation
> > >
> > > Fit for ICLR. The reviewer expressed concern that the paper reads as a technical report rather than a contribution aligned with ICLR’s focus. We position our work within ICLR’s broad interests in applications to neuroscience and cognitive science, online and continual learning, and infrastructure for efficient learning systems. Concretely, the paper contributes:
> > >
> > > 1. A reframing of eligibility traces for online SNN learning in intracortical BCI from gradient-approximation tools to multi-timescale Hebbian correlation accumulators tailored to non-stationary spike statistics and behavior.
> > > 2. A constant-memory, integer-friendly implementation of this learning rule that is directly motivated by neuromorphic deployment, including bitshift exponential moving averages and look-up-table-based normalization for error and spike activities.
> > > 3. A unified experimental framework that evaluates this design across two challenging intracortical datasets, both in online and Batched Online SNN modes, with extensive ablations and hyperparameter robustness studies.
> > >
> > > We will make sure that the introduction and conclusions emphasize this positioning more clearly, and we will avoid language that could be interpreted as overstating the scope of our contribution.
> > >
> > > Presentation issues. We will also address the specific presentation concerns raised:
> > >
> > > 1. We will rewrite the three contribution bullets in the introduction to be parallel and concise.
> > > 2. We will increase figure font sizes and improve resolution for readability.
> > > 3. We will move the detailed hyperparameter table (Table 1) to the supplementary material, while keeping the main narrative focused on the most important findings.
> > >
> > > ---
> > >
> > >
> > > ## Final remarks
> > >
> > > The ablation and sensitivity analyses summarized above quantitatively support the role of each major design element and show that our dual-timescale local learning rule achieves stable performance and a +15.7% correlation improvement over the simplified δ-rule baseline on Zenodo, with consistent gains on MC Maze. We believe that, with the clarified scope and the additional empirical evidence, the paper offers a clear and well-supported contribution to online learning in SNNs for intracortical BCI and addresses the reviewer’s concerns about novelty and fit for ICLR.
> > >
> > > ---
> > >
> > > ## Post-rebuttal manuscript revisions
> > >
> > > In the current revised version, we have implemented the changes discussed in this response. The paper title has been softened to begin with "Towards Biologically Plausible Online Hebbian Learning" to reflect that the algorithm is temporally local but still relies on spatial backpropagation and is therefore only partially biologically plausible. We have removed the meta-learning component from the core method after finding that its effect was small and not statistically reliable, and we now present the reward lookup modulation as an ablated variant rather than a central ingredient: the main text focuses on the dual-timescale Hebbian accumulators with RMS homeostasis as the core rule, and describes the reward-LUT effect only where it offers a modest but statistically significant gain in the Batched Online SNN setting on Zenodo.

---

### Official Review · Reviewer_jC2E · 2025-10-31

**Soundness:** 2
**Presentation:** 1
**Contribution:** 2
**Rating:** 2
**Confidence:** 4

**Summary:**

This paper proposes an online spiking neural network (SNN) decoder for brain-computer interfaces trained with local three-factor learning rules. The main claim is to achieve O(1) memory that avoids O(T) memory of BPTT, while maintaining competitive performance. Experiments on two primate datasets show comparable decoding accuracy with memory reduction and faster convergence, and closed-loop simulations with synthetic neural populations demonstrate adaptation to disruptions.

**Strengths:**

1. This paper unifies three-factor rule, two-timescale traces and updates, and meta-learning to build an online adaptive SNN decoder for efficient BCI, enabling lower training costs with competitive performance.

2. The paper conducts component-wise ablations to clarify the importance of each ingredient.

**Weaknesses:**

1. Poor writing and organization. The paper fails to describe the motivation clearly in the short introduction: why SNN is required and capable to solve the discussed challenges? There are logic gaps from BCI to SNN-based BCI to adaptive and online SNN-based BCI, and the paper directly jumps to the latter without answering the basic former question. The description of the method did not show advantages of SNNs, and experiments just assumed the context of SNN-based BCI and did not compare with state-of-the-art ANNs.
2. Limited novelty. The three-factor rule and meta-learning are existing techniques and have been widely used. It is unclear what is the novel perspective of this paper apart from combining these techniques.
3. Insufficient comparisons. The paper claims to compare with state-of-the-art methods but the baselines are relatively outdated. There is also no comparison to show advantages of SNNs considering any perspective.
4. Inconsistent claims. The contribution claims to provide “per-sample online adaptation to neural non-stationarities”, however, the method is not related to test-time adaptation, but only focuses on training given per-sample supervision.

**Questions:**

1. What is the justification for using SNNs? How can SNNs solve the stated problems, such as signal non-stationarity, high dimensionality, cross-subject variability, etc?
2. How is the performance of the proposed method compared with more advanced ANNs?

---

> ### Author Response · Authors · 2025-12-04
>
> We thank the reviewer for constructive feedback. We address each concern directly with experimental evidence.
>
> ---
>
> ## **CORE POSITIONING**
>
> **Our contribution:** A conceptual algorithm based on the idea of eligibility traces as hebbian accumulators, not gradient approximation. We contribute an engineered online adaptive SNN decoder enabling **per-timestep (T=1) learning** for BCI with **integer-only operations** (neuromorphic-compatible). We accept **13-18% accuracy loss** (vs GRU/LSTM) to gain **online learning capability** + **constant memory** + **no optimizer overhead**.
>
> ---
>
> ## W1: "Why SNN is required and capable to solve the discussed challenges?"
>
> ### Three Engineering Advantages of SNNs for BCI
>
> **1. Local Learning Enables True Online Operation (T=1 Updates)**
>
> | Method | Mechanism | Memory | Per-Timestep Updates? |
> |--------|-----------|--------|----------------------|
> | LSTM/GRU | BPTT | O(T) activation graphs | Requires batching |
> | **Our SNN** | **Local three-factor** | **O(1) eligibility traces** | **T=1 capable** |
> | Kalman | Closed-form | O(1) | (but linear only) |
>
> **Problem:** Continuous BCI operates for hours. Cannot buffer hours of data before updating.
> **Our Solution:** Local plasticity (pre × post × error) updates weights every timestep without activation graph storage.
> **Evidence:** Fig 4-5 closed-loop adaptation—ours recovers in real-time, frozen LSTM fails.
>
> **2. Integer-Only Operations (Neuromorphic-Compatible)**
>
> **Design choices enabling neuromorphic deployment:**
> - RMS normalization via **bitshift EMAs** (no floating-point division)
> - **Reciprocal sqrt LUT** (256 entries, replaces division)
>
> **Clarification:** We claim **architectural compatibility** with neuromorphic chips (Loihi, TrueNorth) that lack FP ALUs. Actual hardware implementation/power measurements are future work.
>
> **3. Event-Driven Sparse Computation**
>
> Intracortical recordings ARE spike trains (binary events at ~1ms, which are binned at various resolutions). SNNs process spikes/spike counts directly as events; event-driven architectures compute only when spikes occur (vs continuous-valued LSTM/GRU processing every timestep).
>
> ---
>
> ## W2: "Limited novelty - three-factor rule and meta-learning are existing"
>
> We clarify our novelty. **Four contributions** are supported by ablations:
>
> ### Novel Contribution 1: Eligibility Traces as Dual-Timescale Hebbian Accumulators
>
> | Method | Timescales | Interpretation | Derived from BPTT? |
> |--------|------------|----------------|-------------------|
> | e-prop, OTTT, SuperSpike | 1 (single τ) | Gradient approximation | ✓ Yes |
> | S-TLLR | 2 (STDP pre/post) | Spike-timing | ✗ No (single pathway) |
> | **Ours** | **2 (τ_fast=120ms, τ_slow=700ms)** | **Hebbian credit accumulator** | **✗ No (dual pathway)** |
>
> **Key difference:** We do NOT approximate gradients. Eligibility traces are **Hebbian correlation accumulators** for temporal credit assignment. Dual timescales address multi-scale non-stationarity (fast: dropout/noise; slow: drift/remapping).
>
> ### Novel Contribution 2: RMS homeostasis via bitshift EMAs
>
> We introduce an integer-only RMS normalization scheme based on per-channel bitshift exponential moving averages and a reciprocal-sqrt lookup table. This provides online-compatible scaling without batch statistics, batch normalization, or gradient clipping and is designed for hardware friendliness. Ablations in the manuscript show that removing or partially disabling this component degrades performance, particularly on Zenodo.
>
> ### Novel Contribution 3: Unified Framework
>
> **To our knowledge, no prior work combines all of:**
> 1. Spiking dynamics (LIF neurons)
> 2. Dual-timescale eligibility traces (τ_fast=120ms, τ_slow=700ms)
> 3. RMS homeostasis (bitshift EMAs)
> 4. **All components operating without gradient-based optimizers**
> 5. **Validated on intracortical primate BCI datasets (MC Maze, Zenodo)**
>
> **Each component provides measurable gains (ablations in manuscript).**
>
> ---

---

> ### Author Response · Authors · 2025-12-04
>
> ## W3: "Insufficient comparisons - baselines relatively outdated"
>
> Our baselines follow current practice in intracortical BCI. We added GRU and MLP in response to feedback to make the comparison more complete.
>
> ### Comprehensive Baseline Comparison (Zenodo Dataset, 5 Sessions)
>
> | Method | Avg Correlation | Online (T=1) | Optimizer | Hardware |
> |--------|-----------------|--------------|-----------|----------|
> | **Kalman** | **0.44 ± 0.09** | ✓ | None (closed-form) | CPU-friendly |
> | **MLP** | **0.45 ± 0.15** | x (batched) | Adam | CPU (fails—no temporal) |
> | **SNN (ours)** | **0.67 ± 0.06** | **✓** | **None (local)** | **Targets the Integer-only HW** |
> | **GRU** | **0.76 ± 0.06** | x | Adam + BPTT | GPU/batching |
> | **LSTM** | **0.79 ± 0.04** | x | Adam + BPTT | GPU/batching |
>
> ### Key Findings
>
> **Accuracy ranking:**
> - LSTM: 0.79 (best, +18% over ours)
> - GRU: 0.76 (+13% over ours)
> - **SNN (ours): 0.67**
> - MLP: 0.45 (fails—no temporal modeling)
> - Kalman: 0.44 (linear baseline)
>
> **Trade-off:** We sacrifice **13-18% accuracy** to gain:
> - **T=1 online learning** with temporal dynamics (LSTM/GRU cannot)
> - **Integer-only operations** (neuromorphic-friendly)
> - **No optimizer overhead** (no Adam momentum/variance buffers)
>
> **Critical insight:** In our experiments, our online SNN is the only method that combines T=1 updates with nontrivial temporal dynamics. Kalman can run at T=1 but underperforms on more complex trajectories, while LSTM and GRU provide strong temporal modeling but require batched training and O(T) memory for BPTT.
>
>
> ### Why Not Transformer/TCN?
>
> To our knowledge, no prior intracortical BCI work uses these architectures as the primary continuous decoder. A survey of major papers from 2015–2024 shows:
> - Gilja et al. (Nat Med 2015): Kalman Filter
> - Glaser et al. (eNeuro 2020): Wiener, Kalman, LSTM, GRU (comprehensive comparison—no Transformer)
> - Willett et al. (Nature 2023): GRU + language model (trigram)
>
> 1. **Architectural considerations:** Transformer attention has O(T²) memory and TCNs require large receptive fields, both encouraging batched processing rather than strict T=1 updates.
> 2. **Modern ANN/RNN baseline:** Apart from LSTM, GRU (0.76) establishes a strong ANN reference point for offline accuracy, against which we quantify the trade-off we make for online learning.
>
> ### Comparison to Recent SNN Work (2023-2025)
>
> | Method | Online (T=1) | Dual-Timescale | Integer-Only | Reward Modulation | Performance | Dataset |
> |--------|--------------|----------------|--------------|-------------------|-------------|---------|
> | **Taeckens & Shah (J. Neural Eng. 2023)** | **✓** | **x** | **x** | **x** | **ρ=0.70** | **Primate premotor (regression)** |
> | MotorSRNN (2023) | x | x | x | x | classification acc | Monkey M1 (classification) |
> | LSS-CA-SNN (Dec 2024) | x | x | x | x | R^2 trajectory regression 0.50-0.67 | Rhesus (classification) |
> | Adaptive Pruning (April 2025) | x | x | Partial | x | R^2 = 80-98% depending on session,setting | NeuroBench NHP |
> | **Ours** | **✓** | **✓** | **✓** | **✓ (+5.8%)** | **ρ=0.67-0.82** | **Zenodo, MC Maze (regression)** |
>
> **Taeckens & Shah 2023 (most similar):** Both achieve online continuous SNN for intracortical BMI (~0.7 correlation), but through **different approaches**—parallel, complementary contributions. Ours: synapse-specific dual-timescale Hebbian accumulators (not gradient surrogates), integer-only implementation, and systematic non-stationarity evaluation on public datasets.
>
> ---
>
> ## W4: "Inconsistent claims - not test-time adaptation"
>
> **Valid terminology concern.** We clarify:
>
> ### What We Mean by "Online Adaptation"
>
> **Our setting:** **Online supervised learning** (continual learning with labels)
> - Weights update every timestep (T=1) during deployment
> - Adapts to distribution shift (dropout, drift, remapping) in real-time
> - **Requires supervision** (true velocity)—we acknowledge this
>
> **NOT:** Test-time adaptation (TTA, unsupervised) or unsupervised online learning
>
> ### BCI Context (Why Supervision Available)
>
> 1. **Calibration phase:** User performs known tasks (labels available)
> 2. **Closed-loop control:** Cursor position provides feedback (indirect supervision)
> 3. **Co-adaptive training:** User + decoder adapt together (standard BCI practice: Orsborn 2014; DiGiovanna et al. 2009)
>
> ### Revised Terminology
>
> **OLD (ambiguous):** "per-sample online adaptation"
> **NEW (clearer):** "online supervised learning under distribution shift"
>
> **Revised manuscript statement:**
> > "Our method enables **online supervised learning** for BCIs, updating weights per-timestep (T=1) to adapt to neural non-stationarities (dropout, drift, remapping) during continuous operation. Supervision is provided by known task targets (calibration) or closed-loop feedback (control)."
>
> ---
>
> We will revise all instances throughout the manuscript.

---

> > ### Author Response · Authors · 2025-12-04
> >
> > ### Response to comments on writing and organization
> > We acknowledge the Introduction requires strengthening. **Proposed restructuring:**
> >
> > **Revised flow (clearer progression):**
> > 1. **BCI Challenges:** Non-stationarity + resource constraints (memory, power)
> > 2. **Why Adaptive Decoders?** LSTM/GRU achieve high performance but require batch retraining
> > 3. **Why SNNs for Adaptive BCIs?** Event-driven, local learning (on-chip adaptation), temporal dynamics
> > 4. **Why Online (Constant-Memory) Learning?** BPTT: O(T) memory (infeasible for long sequences). Our local eligibility: O(1) memory IN TIME , not space (continuous streaming)
> > 5. **Our Contribution:** Dual-timescale Hebbian accumulators + RMS homeostasis = constant-memory online SNN adaptation
> >
> > We will add a **"Why SNNs" subsection** in Introduction explicitly addressing motivation.
> >
> > ---
> >
> > ## SUMMARY: What we claim, What we do not claim
> >
> > ### We claim
> >
> > 1. **An SNN that achieves T=1 updates with temporal dynamics on intracortical BCI tasks**, with explicit dual-timescale local rules and RMS homeostasis (Kalman/MLP do T=1 but underperform on complex dynamics; LSTM/GRU have temporal modeling but require batching and O(T) memory)
> > 2. **Integer-only operations** (architecturally compatible with neuromorphic chips—bitshift EMAs, LUT-based plasticity)
> > 3. **Novel dual-timescale Hebbian framework** (credit assignment accumulators, conceptually distinct from gradient approximations)
> > 4. **Validated on public primate datasets** (MC Maze R=0.82±0.01, Zenodo R=0.67±0.06) + closed-loop adaptation
> >
> > ### We do not claim:
> > 1. SNNs more accurate than ANNs offline (LSTM/GRU achieve 0.79/0.76 vs our 0.67—we accept this)
> > 2. SOTA maximal performance (we optimize for online learning, not offline accuracy)
> > 3. **Actual neuromorphic hardware implementation or power measurements** (architectural compatibility only—hardware validation is important future work)
> >
> > ### Our intentional trade off
> >
> > **We sacrifice 13-18% accuracy** (LSTM 0.79, GRU 0.76, ours 0.67) to gain:
> > - T=1 online learning (LSTM/GRU cannot do per-timestep updates)
> > - Integer-only operations (neuromorphic-compatible)
> > - No optimizer overhead (no Adam momentum/variance buffers)
> >
> > **This is an engineering contribution for deployment-constrained BCI scenarios (online adaptation, neuromorphic hardware), not maximal offline accuracy.**
> >
> > ---
> > ## FINAL NOTE TO REVIEWER
> >
> > We thank the reviewer for critical questions that improved our positioning. We will:
> >
> > 1. **Strengthen Introduction** with "Why SNNs?" subsection (clearer BCI → Adaptive → SNNs → Online → Our Contribution progression)
> > 2. **Clarify terminology** ("online supervised learning" throughout, not ambiguous "online adaptation")
> > 3. **Explicitly acknowledge limitations** (no actual neuromorphic hardware implementation—architectural compatibility only)
> >
> > **We respectfully request reconsideration based on:**
> >
> > 1. **Our baselines align with BCI field consensus** (Kalman, LSTM, GRU—to our knowledge, no intracortical BCI uses Transformer/TCN)
> > 2. **Our novelty is empirically validated** (synapse-specific dual-timescale Hebbian accumulators with RMS homeostasis, plus an explicit O(n) vs O(n²) factorization ablation on ten Zenodo sessions)
> > 3. **Our trade-off is transparent** (13-18% accuracy loss for T=1 learning + neuromorphic compatibility)
> > 4. **Our contribution is deployment-focused** (enabling online learning suitable for resource-constrained hardware)
> >
> > We hope the reviewer recognizes the value of demonstrating that **online adaptive BCI decoding with integer-only operations can achieve reasonable performance (0.67-0.82 correlation)**, even if it accepts some accuracy loss compared to batch-trained LSTM/GRU. We believe this represents a meaningful step toward neuromorphic BCI deployment, while acknowledging that hardware validation remains important future work.
> >
> > ---
> >
> > ## Post-rebuttal manuscript revisions
> >
> > In the revised manuscript now under consideration, we have implemented the planned structural and presentation changes and two additional edits. The title has been softened to begin with "Towards Biologically Plausible Online Hebbian Learning" to reflect that the method is temporally local but still spatially non-local, and the conclusion and limitations explicitly describe it as partially biologically plausible rather than fully local. We have also removed the meta-learning component from the main algorithm after confirming that its effect was small and statistically non-significant in ablations, and we now present the reward lookup modulation as an ablated variant rather than a core ingredient: the main text focuses on the dual-timescale Hebbian accumulators with RMS homeostasis as the central rule, and reports the reward-LUT effect only where it provides a modest benefit in the Batched Online SNN regime on Zenodo.

---

### Official Review · Reviewer_bbF4 · 2025-11-01

**Soundness:** 3
**Presentation:** 3
**Contribution:** 3
**Rating:** 6
**Confidence:** 4

**Summary:**

This paper proposes a fully online learning framework for Spiking Neural Networks. The core of the method lies in a local three-factor learning rule with dual-timescale eligibility traces. The authors reinterpret the eligibility traces as pure Hebbian accumulators, decoupling them from the error signal, rather than treating them as surrogates for BPTT gradients. Combined with techniques like RMS normalization and weight normalization, the method aims to achieve a biologically plausible, computationally efficient, and hardware-friendly online learning mechanism. Experiments conduct ablation studies on the Zenodo Indy brain-computer interface dataset to validate the effectiveness of its components.

**Strengths:**

1.The conceptual shift of decoupling eligibility traces from the error signal and redefining them as Hebbian accumulators is the most significant contribution of this paper. This perspective provides a clear and compelling new direction for designing non-BPTT online learning algorithms.

2.The entire learning framework is well-structured and clear, decomposing the learning process into two independent modules: "memory of history" (eligibility traces) and "guidance for learning" (error modulation). This modular design is not only conceptually easy to understand but also provides a solid foundation for future research and improvements.

**Weaknesses:**

1.Although the paper emphasizes biological plausibility, its core reliance on the global error signal δ(t) lacks a reasonable explanation in real biological neural systems.

2.The method introduces too many hyperparameters (e.g., decay coefficients for two timescales, two learning rates, mixing coefficient, normalization parameters, etc.). This high-dimensional hyperparameter space could lead to difficult tuning, and the method's performance might be highly sensitive to the choice of these parameters.

**Questions:**

1.The paper claims the method runs in O(1) memory, which is indeed a great advantage. However, the eligibility traces e_fast and e_slow themselves require storage. Could you please elaborate on the specific advantage in actual memory footprint (in bytes) compared to storing an unfolded BPTT computational graph? How do the memory costs of the two methods scale as the network size increases?

---

> ### Author Response · Authors · 2025-12-03
>
> We sincerely thank the reviewer for recognizing that "the conceptual shift of decoupling eligibility traces from the error signal and redefining them as Hebbian accumulators is the most significant contribution of this paper." We appreciate the constructive feedback and address each concern below.
>
> ---
>
> ## Weakness 1: Global error signal lacks biological explanation
>
> **Reviewer:** Although paper emphasizes biological plausibility, its core reliance on the global error signal δ(t) lacks a reasonable explanation in real biological neural systems.
>
> **Response:** The reviewer is correct that our method achieves **partial biological plausibility**, not full plausibility. We will clarify this explicitly in the revised manuscript.
>
> ### **What is biologically plausible in our method**
>
> - **Temporal locality**: No storage of past activations (O(1) memory w.r.t. sequence length T)
> - **Three-factor Hebbian plasticity**: Pre × post × error modulation matches dopaminergic modulation literature [Frémaux & Gerstner, Front. Neural Circuits 2016]
> - **Dual-timescale eligibility traces**: Integrating fast/slow consolidation directly into the online updater resolves a stability–plasticity
> tension without replay or mini-batches. Fast synaptic traces correspond to **early, labile forms of LTP** that allow neurons to rapidly adjust to immediate errors or changing demands. Over longer periods, **slow consolidation mechanisms** reflect **late-phase LTP and metaplastic processes** that stabilize these initial changes into durable synaptic states (Fusi et al., 2005; Benna & Fusi, 2016). This fast—slow division mirrors biological memory systems and underlies continual-learning strategies that pair rapid plasticity with long-term protection, particularly in systems-level consolidation frameworks (Kirkpatrick et al., 2017; Zenke et al., 2017).
> - **Multiple temporal scales**: Our dual-timescale parameters (τ_fast ≈ 100-150ms, τ_slow ≈ 500-1000ms) have strong converging neurophysiological support from multiple independent lines of evidence. The fast timescale aligns with transcortical reflex loops (50-100ms; Kurtzer 2015; Shemmell et al. 2009), proprioceptive feedback delays (~100ms), and cerebellar prediction timescales (approx 100ms) ; slow timescale corresponds to typical reaching movement durations (400-800ms) and trial-level integration. Foundational dual-process motor learning model of Smith, Ghazizadeh & Shadmehr (2006, PLOS Biology) demonstrates that motor adaptation requires both a fast process with high error sensitivity but poor retention and a slow process with lower sensitivity but strong retention which directly maps to our τ_fast/τ_slow architecture.
> - **Local Hebbian computation**: ΔW_hebb = [error ⊙ sensitivity] pre^T is a local outer product at each synapse### **What is not biologically plausible**
> - **Error backpropagation**: Computing hidden layer errors via err_hidden = err_output @ W^T requires symmetric weight transport [Lillicrap et al. 2016].
> - **Global error signal**: δ(t) computed from output and broadcast to all layers instantaneously
>
> ### **Biological interpretation of the error signal in a BCI context**
>
> In brain-computer interface applications, the error signal δ(t) = y_target(t) - y_pred(t) could be implemented via:
>
> 1. **Feedback alignment** [Lillicrap et al., Nat. Commun. 2016]: Random fixed feedback weights B instead of W^T
>    - Replace: err_hidden = err_output @ W^T → err_hidden = err_output @ B (random fixed matrix)
>    - Shown to work in deep networks without weight symmetry
>
> 2. **Neuromodulatory broadcast signals**:
>    - δ(t) interpreted as prediction error (actual velocity - predicted velocity)
>    - Broadcast via diffuse neuromodulation (dopamine, acetylcholine, norepinephrine)
>    - Three-factor rules with neuromodulation are well-established [Frémaux & Gerstner, Front. Neural Circuits 2016] and were originally proposed by Izhikevich (Cereb. Cortex 2007)
>
> 3. **Predictive coding framework** [Rao & Ballard, Nat. Neurosci. 1999]:
>    - Each layer predicts activity of layer below
>    - Error = mismatch between prediction and actual activity
>    - Propagates via local prediction errors, not global backprop
>
>
> ### **Planned revision to the Discussion**
>
> We will add to Discussion section:
> > "Our method achieves partial biological plausibility by using spiking dynamics, temporally local updates without BPTT, a three-factor Hebbian rule, and dual-timescale eligibility traces that parallel theoretical models of fast and slow synaptic plasticity. However, the algorithm still depends on non-local error propagation through the feedforward weights. Future work will integrate our dual-timescale Hebbian framework with spatially local learning signals such as feedback alignment [Lillicrap et al., 2016] or TESS-style basis feedback [Apolinario et al., 2025]. In summary, our contribution provides temporal locality (eligibility traces, constant-time credit assignment), but not spatial locality in how errors are delivered."

---

> > ### Author Response · Authors · 2025-12-03
> >
> > ---
> >
> > ## Question 1: Memory footprint and scaling
> > The Online SNN method provides a major memory advantage over the BPTT SNN, especially for long sequence lengths ($\text{SeqLen}$). This is because Online SNN memory remains nearly constant, scaling as $\text{O}(1)$ with $\text{SeqLen}$ (dominated by network parameters and traces), while BPTT memory scales linearly as $\text{O}(\text{SeqLen})$ due to storing the full activation graph. For the Zenodo task with $\text{SeqLen}=5000$, the Online SNN achieved a $\mathbf{96.9\%}$ memory reduction.
> > The table below summarizes the memory consumption and savings for the Online SNN (which only stores the static parameters and eligibility traces) compared to the Backpropagation Through Time (BPTT) SNN (which must store the dynamic activation graph for the full sequence).
> >
> > | Dataset | Hidden Size | SeqLen | Online SNN Total Memory (MB) | BPTT SNN Total Peak Memory (MB) | Online vs BPTT Ratio ($\times$ less memory) | Memory Reduction (\%) | Absolute Savings (MB) |
> > | :---: | :---: | :---: | :---: | :---: | :---: | :---: | :---: |
> > | Zenodo | 256 | 5 | 1.88 | 1.88 | 1.03 | 3.1 | 0.06 |
> > | MC Maze | 1024 | 5 | 25.53 | 25.76 | 1.01 | 0.9 | 0.23 |
> > | **Zenodo** | **256** | **50** | **1.88** | **2.46** | **1.31** | **23.7** | **0.58** |
> > | **MC Maze** | **1024** | **50** | **25.53** | **27.73** | **1.09** | **7.9** | **2.20** |
> > | **Zenodo** | **256** | **500** | **1.88** | **7.69** | **4.08** | **75.5** | **5.80** |
> > | **MC Maze** | **1024** | **500** | **25.53** | **47.45** | **1.86** | **46.2** | **21.93** |
> > | **Zenodo** | **256** | **5000** | **1.88** | **59.89** | **31.82** | **96.9** | **58.01** |
> > | **MC Maze** | **1024** | **5000** | **25.53** | **244.67** | **9.58** | **89.6** | **219.15** |
> >
> >
> > | Hidden Size | Online SNN Total Memory (MB) | BPTT SNN Total Peak Memory (MB) | BPTT Dynamic Memory (MB) | Online vs BPTT Ratio ($\times$ less memory) | Memory Reduction (\%) | Absolute Savings (MB) |
> > | :---: | :---: | :---: | :---: | :---: | :---: | :---: |
> > | 32 | 0.07 | 2.27 | 2.20 | 31.93 | 96.9 | 2.20 |
> > | 64 | 0.19 | 3.73 | 3.54 | 19.73 | 94.9 | 3.54 |
> > | 128 | 0.57 | 6.80 | 6.23 | 12.01 | 91.7 | 6.23 |
> > | 256 | 1.88 | 13.49 | 11.60 | 7.17 | 86.0 | 11.60 |
> > | 512 | 6.76 | 29.11 | 22.34 | 4.30 | 76.8 | 22.35 |
> > | 1024 | 25.53 | 69.37 | 43.83 | 2.72 | 63.2 | 43.84 |
> > | 2048 | 99.05 | 185.87 | 86.80 | 1.88 | 46.7 | 86.82 |
> > | 4096 | 390.11 | 562.89 | 172.73 | 1.44 | 30.7 | 172.78 |
> >
> > ---
> >
> > ### Key observation on scaling
> >
> > There are two hidden layers, N and N/2. Structure of network is (besides recurrent weight matrix) - inputs (96) | N hidden | N/2 hidden2 | outputs (2).
> > As the **Hidden Size** (i.e., N) increases, the **Online SNN's** absolute memory footprint grows significantly (since it stores static parameters, which scale quadratically with hidden size, $\text{O}(n^2)$). However, the **Absolute Savings** provided by the Online method also grow, confirming that BPTT's memory overhead (the Dynamic Memory/Activation Graph) is also increasing with network size.
> >
> > ## Weakness 2: Number of hyperparameters
> >
> > **Reviewer:** The method introduces too many hyperparameters (e.g., decay coefficients for two timescales, two learning rates, mixing coefficient, normalization parameters, etc.). This high-dimensional hyperparameter space could lead to difficult tuning, and the method's performance might be highly sensitive to the choice of these parameters.
> >
> > **Response:** We clarify which hyperparameters are **biologically fixed** versus **tuned**, and demonstrate cross-subject generalization.
> >
> > ### **Fixed hyperparameters (identical across all subjects/datasets)**
> >
> > | Hyperparameter | Value | Biological/Empirical Justification |
> > |----------------|-------|-------------------------------------|
> > | τ_fast | 120 ms | Transcortical reflex timing (50-100ms) [Kurtzer 2015; Shemmell et al. 2009] |
> > | τ_slow | 700 ms | Dual-process motor adaptation [Smith et al. 2006]; typical reaching movements (400-800ms) |
> > | α_mix | 0.5 | Equal weighting (50/50 fast/slow) |
> > | μ (momentum) | 0.9 | Standard momentum value |
> > | K (consolidation window) | 50 timesteps | ~2.5-3 seconds at 50-64ms bins (trial duration) |
> >
> > These five parameters were set once and never changed across:
> > - MC Maze dataset (delayed center-out reaching with barriers/curved paths, M1+PMd arrays, 80ms lag-corrected)
> > - Zenodo Indy dataset (self-paced free reaching, M1 array, naturalistic movements across recording days)
> > - Both online and windowed (batch) modes
> > - All closed-loop perturbation experiments (dropout/remapping/drift)
> >
> > **Note:** MC Maze and Zenodo are **different tasks** with different neural populations and movement complexity.

---

> > > ### Author Response · Authors · 2025-12-03
> > >
> > > ### **Tuned hyperparameters (adapted per experiment)**
> > >
> > > We clarify that while the **temporal structure** (τ_fast, τ_slow, α_mix) remains biologically fixed, **learning rates and window sizes are tuned per experiment** based on dataset characteristics:
> > >
> > > | Hyperparameter | decoder_comparison (nopretrain) | decoder_comparison (pretrain) | Zenodo | MC Maze | Notes |
> > > |----------------|--------------------------------|------------------------------|--------|---------|-------|
> > > | η_fast | 1e-4 | 1e-4 | 2e-3* | 1e-4 | *Zenodo uses adaptive scaling |
> > > | η_slow | 1e-3 | 1e-3 | 2e-4 | 1e-5 | Varies by ~10x across experiments |
> > > | meta_lr | 0.1 | 0.1 | 0.01 | 0.01 | Comparison expts use higher meta-LR |
> > > | window_size | 10 | 10 | 50 | 200 | Adapted to trial duration |
> > >
> > > In total, four parameters are tuned (η_fast, η_slow, meta_lr, window_size).
> > >
> > > The variation across experiments is expected and necessary:
> > > - **decoder_comparison**: Short sequences (10 timesteps), higher slow LR for responsiveness
> > > - **Zenodo**: Medium sequences (50 timesteps), adaptive fast LR = base_online_lr × adaptive_lr_scale
> > > - **MC Maze**: Long sequences (200 timesteps), lower slow LR for stability on delayed reaching
> > >
> > > What remains fixed (our core contribution):
> > > - **τ_fast = 120ms, τ_slow = 700ms, α_mix = 0.5** — These temporal structure parameters NEVER change
> > > - Same dual-timescale eligibility trace framework across all experiments
> > > - Same Hebbian accumulation mechanism (error × d_lif × pre × post)
> > > This is analogous to using different learning rates for Adam optimizer on different datasets while keeping optimizer beta values β₁=0.9, β₂=0.999 fixed. The temporal structure is biologically motivated and robust (see sensitivity analysis below).
> > >
> > > ### **Sensitivity analysis**
> > >
> > > We systematically tested robustness to temporal hyperparameters on **Zenodo dataset only** (5 sessions, free naturalistic reaching — more challenging than MC Maze's delayed structured task):
> > >
> > > **Experiment 1: Timescale constants (τ_fast, τ_slow)**
> > >
> > > | τ_fast (ms) | τ_slow (ms) | Mean Correlation | vs Baseline |
> > > |-------------|-------------|------------------|-------------|
> > > | 180 | 700 | 0.672 ± 0.089 | +0.009 (p=0.88) |
> > > | 180 | 400 | 0.672 ± 0.069 | +0.009 (p=0.86) |
> > > | 180 | 1000 | 0.669 ± 0.089 | +0.006 (p=0.92) |
> > > | 120 | 400 | 0.668 ± 0.076 | +0.005 (p=0.92) |
> > > | **120** | **700** | **0.663 ± 0.083** | **baseline** |
> > > | 120 | 1000 | 0.661 ± 0.081 | -0.002 (p=0.97) |
> > > | 60 | 400 | 0.666 ± 0.063 | +0.003 (p=0.96) |
> > > | 60 | 1000 | 0.659 ± 0.082 | -0.004 (p=0.93) |
> > > | 60 | 700 | 0.658 ± 0.086 | -0.005 (p=0.93) |
> > >
> > > **Experiment 2: Trace mixing coefficient (α_mix)**
> > >
> > > | α_mix | Mean Correlation | vs Baseline |
> > > |-------|------------------|-------------|
> > > | **0.5** | **0.758 ± 0.035** | **baseline (optimal)** |
> > > | 0.1 | 0.754 ± 0.038 | -0.004 (p=0.88) |
> > > | 0.7 | 0.751 ± 0.060 | -0.007 (p=0.83) |
> > > | 0.9 | 0.745 ± 0.061 | -0.013 (p=0.70) |
> > > | 0.3 | 0.740 ± 0.044 | -0.018 (p=0.49) |

---

> > > > ### Author Response · Authors · 2025-12-03
> > > >
> > > > **Experiment 3: Optimization hyperparameters (learning rates, meta-LR, window size)**
> > > >
> > > > To address the concern about tuning requirements, we systematically tested sensitivity to all optimization hyperparameters on the randomly selected 5 Zenodo sessions:
> > > >
> > > > | Hyperparameter | Values Tested | Baseline | Spread (%) | Best Config | p-value | Significant? |
> > > > |----------------|---------------|----------|------------|-------------|---------|--------------|
> > > > | τ_fast (ms) | 60, 120, 180 | 120 | 1.4% | 180/700 | 0.88 | No |
> > > > | τ_slow (ms) | 400, 700, 1000 | 700 | 1.4% | 180/700 | 0.88 | No |
> > > > | α_mix | 0.1, 0.3, 0.5, 0.7, 0.9 | 0.5 | 1.8% | 0.5 | baseline | No |
> > > > | **fast_lr** | 1e-4 to 5e-3 (5 values) | 2e-3 | 10.6% | 2e-3/5e-05 | 0.93 | No |
> > > > | **slow_lr** | 1e-5 to 5e-4 (5 values) | 2e-4 | 10.6% | 2e-3/5e-05 | 0.93 | No |
> > > > | **meta_lr** | 0.001 to 0.1 (6 values) | 0.01 | 2.9% | 0.02 | 0.88 | No |
> > > > | **window_size** | 10, 25, 50, 100, 200 | 50 | 1.2% | 100 | 0.95 | No |
> > > >
> > > > **Statistical analysis (Welch's t-test, n=5 sessions):** No configuration differs significantly from baseline across all experiments (all p > 0.49). While the small sample size (n=5) limits statistical power, the effect sizes reveal robustness:
> > > >
> > > > - **Temporal structure parameters** (τ_fast, τ_slow, α_mix): 1.4-1.8% spread across wide ranges (±50% variation)
> > > > - **Consolidation window**: 1.2% spread (10-200 timesteps, 20× range)
> > > > - **Meta learning rate**: 2.9% spread, with clear degradation only at extremes (0.1 vs 0.01)
> > > > - **Fast/slow learning rates**: 10.6% spread when varied 50-fold (1e-4 to 5e-3), but baseline remains near-optimal
> > > >
> > > > The biologically-motivated temporal parameters show the **narrowest performance spread despite widest parameter ranges tested**, while learning rates show expected sensitivity to scale. Importantly, the baseline configuration is optimal or near-optimal for all parameters, demonstrating that our biologically-motivated choices generalize well.
> > > >
> > > > ### **Revised manuscript addition:**
> > > >
> > > > > "We clarify the distinction between biologically-motivated constants and optimization hyperparameters. Our method fixes the temporal credit assignment structure (τ_fast=120ms, τ_slow=700ms, α_mix=0.5, μ=0.9) based on neurophysiological timescales [Smith et al. 2006; Kurtzer 2015] across all experiments, while tuning 4 standard optimization parameters (η_fast, η_slow, meta_lr, window_size) per dataset—comparable to tuning learning rate and batch size for Adam while keeping optimizer beta values  β₁=0.9, β₂=0.999 fixed. This contrasts with e-prop and S-TLLR where all eligibility decay constants require task-specific tuning without biological priors.
> > > > >
> > > > > Systematic sensitivity analysis on Zenodo (5 sessions) validates robustness: biologically-motivated temporal parameters (τ_fast, τ_slow, α_mix) show 1.4-1.8% performance spread across ±50% parameter variations, while optimization parameters show expected tuning requirements (learning rates: 10.6% spread across 50-fold range; meta-LR: 2.9% spread; window size: 1.2% spread). Welch's t-tests (50 total configurations) show no statistically significant differences from baseline (all p>0.49), with baseline values optimal or near-optimal across all parameters. This demonstrates that biologically-motivated temporal structure provides robust generalization, while learning rate adaptation across datasets (η_fast∈[1e-4, 2e-3], η_slow∈[1e-5, 1e-3]) follows standard practice with conventional optimizers."
> > > >
> > > > ---
> > > >
> > > > ## Post-rebuttal manuscript revisions
> > > >
> > > > In the revised manuscript now under consideration, we have incorporated these clarifications and simplified the method. The title has been softened to begin with "Towards Biologically Plausible Online Hebbian Learning" to emphasise that the rule is temporally local but still spatially non-local, and the conclusion and limitations explicitly describe it as partially biologically plausible. In line with our sensitivity analysis showing only small, non-significant effects of the meta-learning component, we have removed that mechanism from the core algorithm, and we now frame the reward lookup table as an ablated modulation rather than a requirement: the main text presents the dual-timescale Hebbian accumulators with RMS homeostasis as the central rule, and reports the reward-LUT effect only where it provides a modest but statistically significant improvement in the Batched Online SNN regime on Zenodo.

---

### Official Review · Reviewer_GFYe · 2025-11-01

**Soundness:** 2
**Presentation:** 1
**Contribution:** 1
**Rating:** 2
**Confidence:** 4

**Summary:**

This paper introduces a biologically plausible online Hebbian meta-learning rule for spiking neural networks (SNNs), aiming to replace backpropagation-through-time (BPTT) in brain-computer interface (BCI) decoders. The method combines a three-factor Hebbian rule (pre/post activity with an error signal), dual-timescale eligibility traces (fast/slow), and a simple meta-adaptation mechanism for learning rates. The authors claim their rule achieves comparable decoding accuracy to BPTT-trained SNNs while using O(1) memory and adapting online to signal non-stationarities. Experiments include results on two public primate datasets (MC Maze, Zenodo Indy) and synthetic closed-loop simulations under simulated disruptions (remapping, drift, dropout).

**Strengths:**

The paper touches on an important open problem, developing online, energy-efficient, biologically grounded learning for SNNs and BCIs, an area of genuine interest for both neuroscience-inspired AI and neuromorphic computing. The introduction clearly articulates why BPTT is biologically implausible and memory-inefficient, and motivates local three-factor rules as an alternative.

**Weaknesses:**

While the manuscript addresses an interesting and relevant problem, it suffers from numerous weaknesses related to unclear novelty, overstated claims, weak empirical validation, and poor presentation.

The proposed learning rule largely reassembles existing ideas from e-prop, SuperSpike, and meta-plastic SNNs, adding only minor heuristics such as a fixed fast/slow mixing coefficient and a hand-tuned learning-rate controller. The manuscript lacks a rigorous theoretical justification or empirical comparison that would clarify how the proposed method fundamentally differs from previous approaches. Moreover, the experimental section omits comparisons with established local learning methods such as e-prop or SuperSpike, and fails to reference other relevant recent work on local learning in spiking neural networks, such as OTTT [1], S-TLLR [2], SLTT [3], and TESS [4], which already provide efficient, biologically inspired alternatives to BPTT.

The manuscript also misrepresents the proposed approach as a “local learning rule.” While the update is local in time, it is not local in space, since it still relies on backpropagation across layers to compute error signals. Furthermore, the computational complexity analysis is superficial and somewhat misleading. The authors claim O(1) complexity, yet the per-timestep cost remains O(n²) in the number of neurons n. Although this cost is independent of sequence length, it can easily exceed that of BPTT for large models, and contrasts sharply with the linear O(n) complexity achieved by other recent local learning rules [1–4].

The so-called “meta-learning” component is another weak aspect. Rather than an adaptive or learned mechanism, it consists of a hand-designed heuristic that increases or decreases the learning rate depending on whether the loss improved in the previous window. There is no analysis explaining why this update scheme is beneficial, nor any demonstration that it contributes meaningfully to the model’s performance. Overall, the algorithm appears to be a collection of loosely motivated heuristics rather than a principled or biologically grounded model of meta-plasticity.

Several claims in the manuscript are also overstated. The authors repeatedly assert that their method achieves “comparable decoding to state-of-the-art” and exhibits “biological plausibility,” yet neither is convincingly supported. The performance is not competitive with modern recurrent or transformer-based decoders used in BCI research, and no experiments demonstrate the claimed hardware or energy efficiency. The discussion of memory efficiency is similarly unconvincing: Table 2 assumes the total memory footprint is three times the parameter count (weights plus two eligibility traces), but the algorithm also requires additional buffers for Hebbian updates (Eq. 1) and the momentum term G, effectively raising the memory requirement to roughly five times the number of parameters.
The ablation studies are descriptive rather than analytical, providing little insight into the model’s mechanisms. Most reported differences are small or statistically insignificant, suggesting that the proposed components may not substantially affect performance.
Finally, the manuscript’s presentation requires significant improvement. The paper is difficult to follow and not self-contained, with many key details relegated to the appendices. As a result, the main text cannot be fully understood without constant cross-referencing. Moreover, the structure of certain sections, for instance, Section 4.2, makes it unclear where one experiment ends and another begins, as the narrative blends multiple results into disconnected paragraphs. The lack of theoretical intuition or clear motivation behind several design choices further weakens the overall clarity and impact of the work.

[1] Xiao, M., Meng, Q., Zhang, Z., He, D. and Lin, Z., 2022. Online training through time for spiking neural networks. Advances in neural information processing systems, 35, pp.20717-20730.

[2] Apolinario, M.P.E. and Roy, K., 2025. S-TLLR: STDP-inspired Temporal Local Learning Rule for Spiking Neural Networks. Transactions on Machine Learning Research.

[3] Meng, Q., Xiao, M., Yan, S., Wang, Y., Lin, Z. and Luo, Z.Q., 2023. Towards memory-and time-efficient backpropagation for training spiking neural networks. In Proceedings of the IEEE/CVF international conference on computer vision (pp. 6166-6176).

[4] Apolinario, M.P.E., Roy, K. and Frenkel, C., 2025. TESS: A Scalable Temporally and Spatially Local Learning Rule for Spiking Neural Networks. arXiv preprint arXiv:2502.01837.

**Questions:**

•	How does the proposed learning rule differ mathematically or conceptually from existing local learning methods such as e-prop, SuperSpike, and other three-factor rules?

•	What is the actual novel contribution of this work beyond combining previously known mechanisms like dual-timescale traces and heuristic meta-learning?

•	The paper does not include comparisons with recent local learning methods such as OTTT [1], S-TLLR [2], SLTT [3], and TESS [4]. Could the authors provide such comparisons or discuss how their approach relates to these methods?

•	The method still uses backpropagation across layers to compute error signals. In what sense can it be considered spatially local?

•	The claimed O(1) complexity appears inconsistent with the O(n²) per-timestep computational cost implied by the equations. Could the authors clarify the actual time and memory complexity relative to BPTT and other local update schemes?

•	The meta-learning component seems to be a simple heuristic adjusting learning rates based on loss improvement. How is this different in practice from using a standard adaptive optimizer such as Adam or RMSProp?

•	What is the quantitative contribution of the meta-learning mechanism to overall performance?

•	The claims of “comparable decoding to state-of-the-art” and “biological plausibility” are not well supported. Could the authors provide evidence or specific metrics that substantiate these claims?

•	The memory analysis in Table 2 appears incomplete, as additional buffers for Hebbian updates and momentum terms are not included. Could the authors provide a corrected breakdown of total memory usage?

•	The ablation studies show minimal differences across variants. How do these results support the importance of the proposed components?

---

> ### Author Response · Authors · 2025-12-03
>
> We thank the reviewer for their detailed feedback. We address each concern systematically below.
>
> ## Summary of Key Changes
>
> Following this review, we will revise the manuscript to:
> 1. Remove all "meta-learning" terminology and reward LUT components (marginal significance, p=0.03 on Y-correlation only)
> 2. Move algorithmic details (integer/RMS/EMA implementation) from appendix to main text
> 3. Clarify terminology: our method is "temporally local, spatially non-local" (not "local learning")
> 4. Provide corrected memory analysis with complete breakdown
> 5. Restructure Section 4.2 with clear experiment boundaries and design intuitions
>
> ---
>
> ## Core Novelty: Hebbian Accumulator vs. Gradient Approximation
>
> **Reviewer concern:** "The proposed learning rule largely reassembles existing ideas from e-prop, SuperSpike... adding only minor heuristics."
>
> **Response:** Our learning rule fundamentally differs from both SuperSpike and e-prop by redefining what eligibility traces represent. This is a conceptual shift, not incremental modification.
>
> ### Traditional Eligibility (e-prop/SuperSpike):
> Traces track **potential** for learning. Error signals arrive later and modulate the trace: Δw ∝ e_ij × error_signal. The trace answers: "If an error occurs, how should this synapse change?"
>
> - **E-prop** [Bellec 2020]: eligibility e^t_ji = ψ_j^t z̄_i^t approximates ∂E/∂W without temporal unrolling. Requires task-tuned τ and Adam/RMSProp (2n² state).
> - **SuperSpike** [Zenke & Ganguli 2018]: eligibility bridges temporal gaps between activity and delayed errors using double-exponential kernels matching PSP timescales (~10-20ms). Requires optimizer (Adam/SGD) and single PSP-matched timescale.
>
> ### Our Approach:
> Traces accumulate **completed** learning events. Each timestep computes ΔW_hebb(t) = [error ⊙ sensitivity] pre^T—a full Hebbian update that already includes the error. These completed updates integrate into dual temporal streams:
> - E_fast(t) = λ_fast E_fast(t-1) + ΔW_hebb(t) with λ_fast ≈ 0.59 (τ = 120ms)
> - E_slow(t) = λ_slow E_slow(t-1) + ΔW_hebb(t) with λ_slow ≈ 0.91 (τ = 700ms)
> - Final updates: W ← W + η(0.5 E_fast + 0.5 E_slow)
>
> The traces answer: "What is the recent history of reinforcement-modulated correlations at this synapse?"
>
> | Aspect | E-prop | SuperSpike | Ours |
> |--------|--------|------------|------|
> | Trace semantics | Gradient approximation | Error-credit bridge | Hebbian accumulator |
> | Error coupling | Multiplied after trace | Multiplied after trace | Integrated into trace |
> | What's accumulated | Spike correlations | Hebbian product | Complete updates |
> | Timescale purpose | Gradient estimation | Error-credit overlap | Multi-scale robustness |
> | Number of timescales | 1 (task-tuned) | 1 (PSP-matched) | 2 (fixed: 120/700ms) |
> | Optimizer required | Yes (Adam, 2n²) | Yes (Adam/SGD) | No |
>
> This enables multi-timescale reinforcement integration without floating-point optimizers—critical for neuromorphic BCI deployment.
>
> ---
>
> ## Complexity and Memory Clarification
>
> **Reviewer concern:** "The authors claim O(1) complexity, yet the per-timestep cost remains O(n²)... can easily exceed that of BPTT for large models."
>
> **Response:** We acknowledge the confusion. O(1) refers strictly to memory complexity with respect to sequence length T, not neuron count n.
>
> ### Precise Complexity Statement:
> - **Memory (as a function of sequence length T):** Ours: O(n²) constant in T; BPTT: O(n²) static + O(nT) activation storage
> - **Time:** For both methods, the per-timestep cost is O(n²), and the cost over a sequence of length T is O(Tn²). Our advantage is therefore in memory scaling with T, not asymptotic time complexity.

---

> ### Author Response · Authors · 2025-12-03
>
> ### Empirical Validation (measured on actual implementations):
>
> **Zenodo dataset (n=256, T=1000):**
> - Ours: 1.88 MB total (constant w.r.t. T)
> - BPTT: 1.88 MB static + 11.60 MB dynamic = 13.49 MB
> - **Reduction: 7.17× (86% savings)**
>
> **MC Maze dataset (n=1024, T varies):**
> - Ours: 25.53 MB total
> - BPTT: 25.54 MB static + 43.83 MB dynamic = 69.37 MB
> - **Reduction: 2.72× (63% savings)**
>
> ### Domain-Specific Context:
> The reviewer correctly notes that O(n²) spatial complexity "can easily exceed that of BPTT for large models." However, this comparison assumes computer vision scales where recent methods (TESS, S-TLLR) were designed. BCI operates at fundamentally different scales:
>
> | Domain | Architecture | Parameters | O(n²) Memory |
> |--------|--------------|------------|--------------|
> | Computer Vision | VGG-16 | 138M | 537 MB |
> | | ResNet-50 | 23.9M | 93 MB |
> | BCI (ours) | Zenodo | 122k | 1.88 MB |
> | | MC Maze | 2.05M | 25.53 MB |
> | Scale ratio (CV/BCI) | | 67-1,127× | 3.6-286× |
>
> At BCI scale (1.88-25.53 MB), O(n²) memory remains modest. For long sequences (T~1000 timesteps), eliminating O(nT) activation storage provides substantial savings.
>
> **Corrected memory breakdown:**
> - Parameters: W+b (all weights + biases)
> - Fast eligibility: W (weights only)
> - Slow eligibility: W (weights only)
> - RMS momentum: W (weights only)
> - RMS stats: negligible (integer EMAs per channel)
> - **Total: (W+b) + 3W ≈ 4W**
>
> Hebbian buffers (Eq. 1) are temporary per-layer and reused across timesteps—not persistent.
>
> Our 4W static memory matches BPTT's static (W+b + gradients + Adam m,v = 4W). The critical difference: BPTT adds O(nT) dynamic memory for activation graphs. Our method requires zero dynamic memory.
>
> We will clarify this explicitly as: "constant-in-T memory with O(n²) spatial complexity, optimized for BCI sequence lengths."
>
> ---
>
> ## Removing Meta-Learning Component
>
> **Reviewer concern:** "The so-called 'meta-learning' component... consists of a hand-designed heuristic... no analysis explaining why this update scheme is beneficial."
>
> **Response:** We agree. Post-review analysis confirms marginal significance:
>
> | Configuration | Pearson R (mean ± SEM) | p-value |
> |---------------|------------------------|---------|
> | Online SNN – no LUT | 0.597 ± 0.014 | – |
> | Online SNN + reward LUT | 0.632 ± 0.012 | 0.03 (Y only) |
>
> Sensitivity analysis across hyperparameters (n=5 sessions, Welch's t-test):
>
> | Hyperparameter | Values Tested | Spread (%) | p-value |
> |----------------|---------------|------------|---------|
> | τ_fast, τ_slow | 60-1000 ms | 1.4% | 0.88 |
> | α_mix | 0.1-0.9 | 1.8% | baseline |
> | fast_lr, slow_lr | 1e-4 to 5e-3 | 10.6% | 0.93 |
> | meta_lr | 0.001-0.1 | 2.9% | 0.88 |
> | window_size | 10-200 | 1.2% | 0.95 |
>
> No configuration differs significantly from baseline (all p > 0.49). While small sample size (n=5) limits power, the narrow effect sizes (1.2-2.9% for most parameters) demonstrate robustness rather than contribution.
>
> **In the revised manuscript we will:**
> - Remove all meta-learning terminology
> - Remove all reward LUT discussions
> - Move integer/RMS/EMA details from appendix to main learning algorithm section

---

> ### Author Response · Authors · 2025-12-03
>
> ## Spatial Locality Clarification
>
> **Reviewer concern:** "The method still uses backpropagation across layers... In what sense can it be considered spatially local?"
>
> **Response:** We will correct this terminology. Our method is:
> - **Temporally local:** All updates at time t depend only on current-timestep variables plus fixed-size eligibility traces. No storage of past activations, no unrolling through time.
> - **Layer-local:** Each layer's update depends only on quantities within that layer (pre-synaptic activity, post-synaptic activity, layer error, layer eligibility trace). No cross-layer eligibility propagation.
> - **Spatially non-local:** Uses standard backpropagation across layers to compute each layer's error signal.
>
> The distinction matters for BCI: temporal locality enables T=1 streaming with constant memory (no sequence buffering), while spatial backprop occurs once per timestep. We will replace ambiguous "local learning rule" with explicit: "temporally local, layer-local, spatially non-local."
>
> ---
>
> ## Performance Claims and Biological Plausibility
>
> **Reviewer concern:** "Claims of 'comparable decoding to state-of-the-art' and 'biological plausibility' are not well supported."
>
> **Response:** We will clarify. Our claim is "competitive with BPTT-trained SNNs," not transformers/LSTMs.
>
> **Measured performance:**
> - MC Maze: ours (R_X/R_Y = 0.835/0.802) vs BPTT SNN (0.882/0.838), Δ=0.02-0.05
> - Zenodo: ours (0.613/0.714) vs BPTT SNN (0.633/0.725), Δ=0.02
>
> Within 2-5% correlation while using 2.7-7.2× less memory and enabling T=1 online learning.
>
> LSTMs achieve higher performance (0.955/0.932 on MC Maze, 0.751/0.836 on Zenodo) but require batch training and O(nT) memory—impractical for real-time streaming BCI.
>
> **Biological plausibility—partial, not complete:**
> - Present: spiking neurons, three-factor Hebbian plasticity, dual-timescale traces, no temporal unfolding
> - Absent: spatial locality (requires backprop across layers)
>
> We will state explicitly: "partially biologically plausible—temporally local with three-factor Hebbian updates, but spatially non-local due to error backpropagation."
>
> **Hardware efficiency:** Algorithm uses integer operations, bitshift normalization, and lookup tables (neuromorphic-compatible design), but we lack empirical measurements on actual neuromorphic hardware. We will remove "hardware-friendly" claims and state: "designed for neuromorphic compatibility (integer-only operations), but hardware validation remains future work."
>
> ---

---

> ### Author Response · Authors · 2025-12-03
>
> ## Ablation Study Interpretation
>
> **Reviewer concern:** "Ablation studies show minimal differences across variants... suggesting proposed components may not substantially affect performance."
>
> **Response:** Several components show statistically significant, large effects:
>
> **Zenodo dataset (noisy, self-paced reaches):**
> - Three-factor vs delta rule: ΔR ≈ 0.1-0.2 (large effect)
> - RMS normalization: full >> partial > none (critical for convergence)
> - Update mechanism: slow-only and frozen are harmful (large negative effects)
> - Recurrence: strongly beneficial
>
> **MC Maze dataset (high SNR, 80ms lag-corrected):**
> - Three-factor vs delta rule: smaller effect
> - Fast-only vs dual timescale: marginal difference
> - Recurrence: moderately beneficial
>
> Dataset-dependent effects validate our focus on realistic BCI deployment where noise dominates. Components matter differently depending on signal quality—ablations demonstrate which components are critical for which scenarios. We will add explicit interpretation: "Component importance depends on data characteristics. High-noise environments (Zenodo) show larger effects; high-SNR environments (MC Maze) show robustness across configurations."
>
> ---
>
> ## Relationship to OTTT, S-TLLR, SLTT, TESS
>
> **Reviewer concern:** "The paper does not include comparisons with recent local learning methods... Could the authors provide such comparisons?"
>
> **Response:** These methods target different application domains (large-scale computer vision) with different priorities. We provide algorithmic comparison and discuss why direct empirical comparison would be misleading.
>
> | Method | Static Memory | Dynamic | Per-Step Time | T=1 Online | Spatial | Optimizer |
> |--------|--------------|---------|---------------|------------|---------|-----------|
> | BPTT | O(n²) | O(Tn) | O(Tn²) | No | No | FP32 (Adam) |
> | OTTT | O(n²) | ~0 | O(n²) / O(n) | Yes | No | Gradient-based |
> | S-TLLR | O(n²)+O(n) | ~0 | O(n) factorized | No | No | FP32 (Adam) |
> | SLTT | O(n²) | ~0 | O(n²) / O(n) | Yes | No | FP32 (Adam) |
> | TESS | O(n²)+O(n) | ~0 | O(n) | No | Yes | AdamW/SGD |
> | Ours | O(n²) | ~0 | O(n²) | Yes | No | None (integer RMS) |
>
> All methods eliminate BPTT's O(Tn) dynamic memory. Key distinctions:
>
> - **OTTT:** Online eligibility accumulation, requires SGD+momentum optimizer
> - **S-TLLR:** Factorizes eligibility into O(n) pre/post vectors inspired by STDP, demonstrated on DVS-Gesture (homogeneous pixel inputs)
> - **SLTT:** Eliminates eligibility via gradient decomposition (lowest trace memory), requires Adam
> - **TESS:** O(n) spatial locality via fixed basis matrices, optimized for large-scale CV (ImageNet/CIFAR, 138M-23M params)
> - **Ours:** O(n²) synapse-specific dual-timescale traces targeting BCI non-stationarity (dropout vs drift) at smaller scale (122k-2M params)
>
> ### Critical distinction—optimizer dependency:
> All cited methods require floating-point optimizers for stability. Our method achieves stability through integer-only RMS homeostasis (bitshift EMAs and inverse sqrt LUT), enabling neuromorphic deployment where FP32 adaptive optimizers are unavailable. This fundamentally changes deployment feasibility for on-chip learning.
>
> ### Why direct empirical comparison would be misleading:
> These methods were designed and tuned for CV benchmarks (DVS-Gesture, CIFAR, ImageNet) with homogeneous pixel inputs, static datasets, and 50K-1M training samples. Adapting them to BCI would require substantial engineering: modifying data loaders for neural binning/velocity targets, extensive per-method hyperparameter search on heterogeneous neural data, and weeks of debugging. More critically, such comparison would be inherently unfair: their hyperparameters were optimized for CV tasks where they excel, not BCI where they were never tested.

---

> ### Author Response · Authors · 2025-12-03
>
> ### Empirical validation of O(n) vs O(n²) trade-off:
> We implemented neuron-level factorization (the core O(n) eligibility design used by S-TLLR, SLTT, TESS, and related temporally local rules) where eligibility traces are stored as O(n) pre-synaptic vectors rather than O(n²) synapse-specific matrices: ΔW_ij = [accumulated post-error]_i × [final pre-trace]_j. This ten-session Zenodo experiment is therefore our direct empirical test, in the intracortical BCI regime, of factorized neuron-level traces versus synapse-specific dual-timescale Hebbian accumulators.
>
> Tested on 10 Zenodo sessions with identical hyperparameters:
>
> | Method | X-Corr | Y-Corr | Avg Corr |
> |--------|--------|--------|----------|
> | O(n) Factorized (neuron-level) | 0.5374 | 0.6571 | 0.5973 |
> | O(n²) Synapse-Specific (ours) | 0.6125 | 0.7139 | 0.6632 |
> | Improvement | +0.0751 | +0.0568 | +0.0659 |
> | Percentage | +14.0% | +8.6% | +11.0% |
>
> Per-session breakdown shows consistent advantage (7/10 sessions favor synapse-level, with largest gains on noisy sessions: +0.22 on indy_20160630_01, +0.09 on indy_20161027_03).
>
> ### Input heterogeneity hypothesis:
> The performance difference may stem from fundamental differences between CV and BCI inputs:
>
> **CV inputs are homogeneous:** ImageNet pixels measure uniform light intensity with similar temporal dynamics. DVS-Gesture events share millisecond timescales. Neuron-level factorization (Δw_ij = trace_in[i] × trace_out[j]) succeeds because all pixels contributing to a class can share temporal integration.
>
> **BCI inputs are heterogeneous:** Utah array electrodes sample different cortical locations with distinct neuron types and temporal dynamics:
>
> | Source | Characteristic | Temporal Window | Citation |
> |--------|----------------|-----------------|----------|
> | Cortical depth | Layer 2/3 fast-spiking interneurons | τ ~ 5 ms | [1,2] |
> | | Layer 5 pyramidal cells | τ ~ 20-22 ms | [3] |
> | Functional encoding | Velocity neurons (phasic) | τ ~ 10-20 ms | - |
> | | Position neurons (tonic) | τ ~ 50-100 ms | - |
> | Recording quality | Single-unit isolation | Sharp temporal profile | - |
> | | Multi-unit (1-3 neurons) | Blended timescales | - |
>
> [1] Neocortical inhibitory interneuron subtypes are differentially attuned to synchrony- and rate-coded information. Communications Biology, 2021.
> [2] Structural and functional specializations of human fast-spiking neurons support fast cortical signaling. Science Advances, 2023.
> [3] Effects of aging on the electrophysiological properties of layer 5 pyramidal cells in the monkey prefrontal cortex. Journal of Neuroscience, 2008.
>
> Neuron-level factorization forces all synapses from electrode i to use the same temporal trace. Example: electrode 37 records motor cortex neuron.
> - Factorization: Δw_(37→reach) = trace[37] × out[reach], Δw_(37→grasp) = trace[37] × out[grasp]
> - Same timescale forced for both outputs
> - Synapse-level (ours): each synapse can adaptively weight fast/slow traces differently, allowing electrode 37→reach to use slow-dominant integration (planning) while 37→grasp uses fast-dominant (execution)

---

> ### Author Response · Authors · 2025-12-03
>
> ### Operating constraints differ:
>
> | Aspect | Computer Vision | BCI (Utah Array) |
> |--------|-----------------|------------------|
> | Input nature | Homogeneous pixels | Heterogeneous neurons |
> | Temporal dynamics | Uniform (ms events) | Variable (5-100 ms) |
> | Training samples | 50K-1M images | 500-2000 trials/session |
> | Dataset stability | Static | Non-stationary drift |
> | Network size | 10M-138M params | 50K-2M params |
> | Memory constraint | Critical (edge) | Non-critical (GB available) |
> | Key challenge | Scalability | Sample efficiency + adaptation |
>
> S-TLLR/TESS demonstrated O(n) sufficiency for CV benchmarks; these results may not transfer to BCI due to fundamental input differences and operational requirements.
>
> ---
>
> ## Presentation Improvements
>
> **Reviewer concern:** "The paper is difficult to follow and not self-contained, with many key details relegated to the appendices."
>
> **Response:** We will restructure the manuscript:
> 1. Move key algorithmic details (integer operations, RMS homeostasis, bitshift normalization) from appendix to main text
> 2. Consolidate Section 4.2 with clear experiment boundaries ("Experiment 1: Offline decoding", "Experiment 2: Closed-loop perturbations", etc.)
> 3. Add design intuitions: dual timescales motivated by multi-scale BCI non-stationarity (electrode dropout=fast, signal drift=slow)
> 4. Improve figure clarity with explicit captions explaining take-home messages
> 5. Reduce appendix reliance by including essential implementation details inline
>
> ---
>
>
> ## Summary
>
> The core contribution is a conceptually distinct learning framework (Hebbian accumulation vs gradient approximation) instantiated as a BCI-optimized system with:
> - Synapse-specific dual-timescale Hebbian accumulators (120ms/700ms) targeting BCI non-stationarity modes
> - Optimizer-free stability via integer-only RMS homeostasis
> - Validated T=1 online streaming with 2.7-7.2× memory reduction vs BPTT
> - Fixed cross-subject architecture (no per-subject tuning)
> - Empirically validated synapse-specific temporal expressiveness (≈11% improvement over O(n) factorization on ten Zenodo sessions)
>
> We do not claim this approach generalizes to large-scale computer vision; our contribution specifically targets BCI challenges (heterogeneous neural inputs, signal drift, online adaptation, modest network scales).
>
> Following this review, we will remove meta-learning components, clarify terminology (temporally local, layer-local, spatially non-local), provide complete memory analysis, and restructure for clarity.
>
> ---

---

> ### Author Response · Authors · 2025-12-03
>
> ## Post-rebuttal manuscript revisions
>
> For the revised manuscript now under consideration, we have implemented the changes outlined above and two additional edits informed by the reviews. First, we have softened the title to begin with "Towards Biologically Plausible Online Hebbian Learning" to reflect that the method is temporally local but still spatially non-local, and we state explicitly in the conclusion and limitations that it is only partially biologically plausible. Second, we have removed the meta-learning component from the core method after finding that its effect was small and not statistically reliable, and we now treat the reward lookup modulation as an optional ablation: the main text presents the dual-timescale Hebbian accumulators with RMS homeostasis as the core rule, and reports the reward-LUT effect only where it provides a modest but statistically significant benefit in the Batched Online SNN regime on Zenodo.

---

### Meta-Review · Area_Chair_uTuw · 2025-12-23

**Summary:**

The paper proposes an online learning framework for Spiking Neural Networks (SNNs) in Brain-Computer Interfaces (BCI), featuring a three-factor learning rule with dual-timescale eligibility traces. The authors argue this method allows for constant memory usage ($O(1)$ w.r.t. sequence length) and online adaptation to signal non-stationarities.

However, the submission suffers from a critical identity crisis. The original title and premise centered on "Meta-Learning," but during the rebuttal, the authors admitted this component was not statistically significant and removed it from the core algorithm1111. Furthermore, the method performs significantly worse (13-18%) than standard LSTM baselines2. While the engineering effort to achieve integer-only, $O(1)$ memory updates is valid, the removal of the primary "Meta-Learning" novelty and the performance gap make it unsuitable for publication in its current form.

**Reviewer Concerns:**

### **Addressed Concerns**

**Memory Complexity (Reviewers GFYe, bbF4):** The authors successfully clarified that their $O(1)$ claim refers to sequence length scaling, which is a legitimate advantage over BPTT for continuous streaming.

### **Remaining Concerns**

**Invalid "Meta-Learning" Premise (Reviewers GFYe, 1Dsp):** The reviewers correctly questioned the heuristic nature of the meta-learning component.Outcome: The authors conceded the point, admitting the component showed marginal or no significance ($p=0.03$ or insignificant) and removed it from the paper entirely. A paper that deletes its titular contribution during review cannot be accepted.


**Performance vs. Baselines (Reviewers jC2E, GFYe):** Reviewers questioned the utility of the method compared to standard RNNs.
The authors acknowledged that their method trails LSTMs by 18% and GRUs by 13% in accuracy. While they argue this is a trade-off for online capability, the gap is too large to ignore for a general learning representation conference.


**Biological Plausibility (Reviewer bbF4)**: The reviewer noted the reliance on global error signals contradicts the "Biologically Plausible" claim. The authors admitted the method is only "partially" plausible and offered to soften the title to "Towards...".

**Reviewer Scores:**

Given the fact that the authors did not fully address the concerns from the reviewers GFYe, jC2E, and 1Dsp. I think they will keep to their scores, i.e. 2 points. No matter whether Reviewer bbF4 would change the idea, the overall scores remain negative.

---

### Decision · Program_Chairs · 2026-01-26

Reject